# Direct ETTIN-auxin interaction controls chromatin states in gynoecium development

André Kuhn[1], Sigurd Ramans Harborough[2], Heather M McLaughlin[1], Bhavani Natarajan[1], Inge Verstraeten[3], Jiří Friml[3], Stefan Kepinski[2], Lars Østergaard[1]*

[1]Department of Crop Genetics, John Innes Centre, Norwich Research Park, Norwich, United Kingdom; [2]Centre for Plant Sciences, Faculty of Biological Sciences, University of Leeds, Leeds, United Kingdom; [3]Institute of Science and Technology, Klosterneuburg, Austria

**Abstract** Hormonal signalling in animals often involves direct transcription factor-hormone interactions that modulate gene expression. In contrast, plant hormone signalling is most commonly based on de-repression via the degradation of transcriptional repressors. Recently, we uncovered a non-canonical signalling mechanism for the plant hormone auxin whereby auxin directly affects the activity of the atypical auxin response factor (ARF), ETTIN towards target genes without the requirement for protein degradation. Here we show that ETTIN directly binds auxin, leading to dissociation from co-repressor proteins of the TOPLESS/TOPLESS-RELATED family followed by histone acetylation and induction of gene expression. This mechanism is reminiscent of animal hormone signalling as it affects the activity towards regulation of target genes and provides the first example of a DNA-bound hormone receptor in plants. Whilst auxin affects canonical ARFs indirectly by facilitating degradation of Aux/IAA repressors, direct ETTIN-auxin interactions allow switching between repressive and de-repressive chromatin states in an instantly-reversible manner.

*For correspondence:
lars.ostergaard@jic.ac.uk

Competing interests: The authors declare that no competing interests exist.

## Introduction

Developmental programmes within multicellular organisms originate from a single cell (*i.e.* a fertilised oocyte) that proliferates into numerous cells ultimately differentiating to make up specialised tissues and organs. Tight temporal and spatial regulation of the genes involved in these processes is essential for proper development of the organism. Changes in gene expression are often controlled by mobile signals that translate positional information into cell type-specific transcriptional outputs (*Hironaka and Morishita, 2012*). In plants, this coordination can be facilitated by phytohormones such as auxin, which controls processes throughout plant development (*Vanneste and Friml, 2009*). In canonical auxin signalling, auxin-responsive genes are repressed when auxin levels are low by Aux/IAA transcriptional repressors that interact with DNA-bound Auxin Response Factors (ARFs). As auxin levels increase, the auxin molecule binds to members of the TIR1/AFB family of auxin co-receptors (*Kepinski and Leyser, 2005*; *Dharmasiri et al., 2005*). The main role of this auxin perception mechanism, besides non-transcriptionally inhibiting growth (*Fendrych et al., 2018*; *Gallei et al., 2019*), is transcriptional reprogramming. This is realised by interaction with Aux/IAA repressors, leading to their ubiquitination and subsequent degradation by the 26S proteasome, ultimately relieving the repression of ARF-targeted loci (*Kelley and Estelle, 2012*; *Leyser, 2018*; *Weijers and Wagner, 2016*).

We recently identified an alternative auxin-signalling mechanism whereby auxin directly affects the activity of a transcription factor (TF) complex towards its downstream targets (*Simonini et al.,*

*2016*; *Simonini et al., 2017*). This mechanism mediates precise polarity switches during organ initiation and patterning and includes the ARF, ETTIN (ETT/ARF3) as a pivotal component. However, ETT is an unusual ARF lacking the Aux/IAA-interacting Phox/Bem1 (PB1) domain (*Simonini et al., 2016*; *Sessions et al., 1997*) and is therefore unlikely to mediate auxin signalling via the canonical pathway. Here, we demonstrate that ETT binds auxin directly and that this interaction determines expression of ETT target genes. In conditions of low auxin levels, ETT interacts with co-repressors of the TOPLESS/TOPLESS-RELATED (TPL/TPR) family to keep chromatin at ETT target loci in a repressed state through HDA19-mediated histone deacetylation. Under high auxin conditions, ETT binds auxin leading to dissociation of TPL/TPR-HDA19 and de-repression of ETT targets.

## Results

ETT can interact with a diverse set of TFs and these interactions are sensitive to the naturally occurring auxin, indole 3-acetic acid (IAA) (*Simonini et al., 2016*). To test if this effect is mediated through a direct interaction between ETT and auxin, we took advantage of the drug affinity-responsive target stability (DARTS) assay (*Lomenick et al., 2011*) and followed the effect of IAA on proteolytic degradation of a FLAG-fused ETT segment conditionally expressed in Arabidopsis (*Figure 1a*). The concentrations of IAA applied (0.1 μM, 1 μM, 10 μM) were selected to saturate the protein with ligand and to ensure maximal protection from proteolysis. As a negative control, benzoic acid (BA) was applied at the same dose as IAA (10 μM and 100 μM) because of its similar size and $pK_a$ as IAA. Treatments with IAA but not BA protected ETT-FLAG from pronase-induced degradation consistent with direct, specific interaction between IAA and ETT in planta.

The region responsible for IAA-sensitivity is situated within the C-terminal part of ETT, known as the ETT-Specific (ES) domain. A protein fragment containing 207 amino acids of the ES domain, $ES^{388-594}$, sufficient for mediating IAA-sensitivity in ETT-protein interactions, was produced recombinantly and shown to be intrinsically disordered (*Simonini et al., 2018*). The sensitivity of ETT-TF interactions to IAA suggests a direct effect of the IAA molecule on the ETT protein. Therefore, to test whether ETT binds IAA, we carried out heteronuclear single quantum coherence (HSQC) nuclear magnetic resonance (NMR) experiments using $^{15}N$-labelled $ES^{388-594}$ protein. The HSQC spectrum, recorded at 5 °C, shows a prominent signal-dense region consistent with the ES domain being largely intrinsically disordered. Interestingly, the spectrum also shows dispersed peaks flanking the signal-dense region indicating that there is nevertheless some propensity to form secondary structure, particularly with a helical character (*Figure 1b*). In addition to this overview of ETT structure, the HSQC NMR probes chemical shifts of protein amide-NH bonds in response to the presence of ligand (*Meyer and Peters, 2003*). We found that a number of residues shifted their position in the spectrum in response to the addition of IAA, whereas addition of BA had no effect (*Figure 1b–d*). These shifts show that certain residues are experiencing a changed chemical environment as a consequence of IAA-binding and this may include the conformational change of a structural motif within the ETT protein. The HSQC experiment therefore demonstrates that ETT binds IAA directly. This experiment has not allowed us to assign signals to specific amino acids and hence there is some uncertainty associated with tracking the chemical shifts of some residues. However, a particularly large change is observed for the tryptophan NH cross peak when IAA is added to the ETT fragment (~10 ppm, rectangle I in *Figure 1b,d*). Since there is only one tryptophan in the ETT fragment used here (W505), this shift can be assigned to this residue and suggests that W505 has a prominent role in the interaction with IAA.

We also used the recombinant ETT fragment in an Isothermal Titration Calorimetry (ITC) assay, which characterises binding of ligands to proteins by determining thermodynamic parameters of the interaction as heat exchange. This experiment revealed interaction between ETT and IAA, while control experiments titrating IAA into buffer without protein and titrating buffer without IAA into the ETT fragment showed no heat exchange (*Figure 1e–g*). We were unable to obtain a $K_d$ value for the interaction, which may be due to the interaction being weak as well as potential protein aggregation. This could reflect that the ETT-auxin interaction is stabilised by interacting protein partners in planta that are not present in this in vitro experiment.

Together, these three independent biochemical methods demonstrate that ETT binds IAA directly thus revealing a key molecular aspect of the non-canonical auxin-signalling pathway.

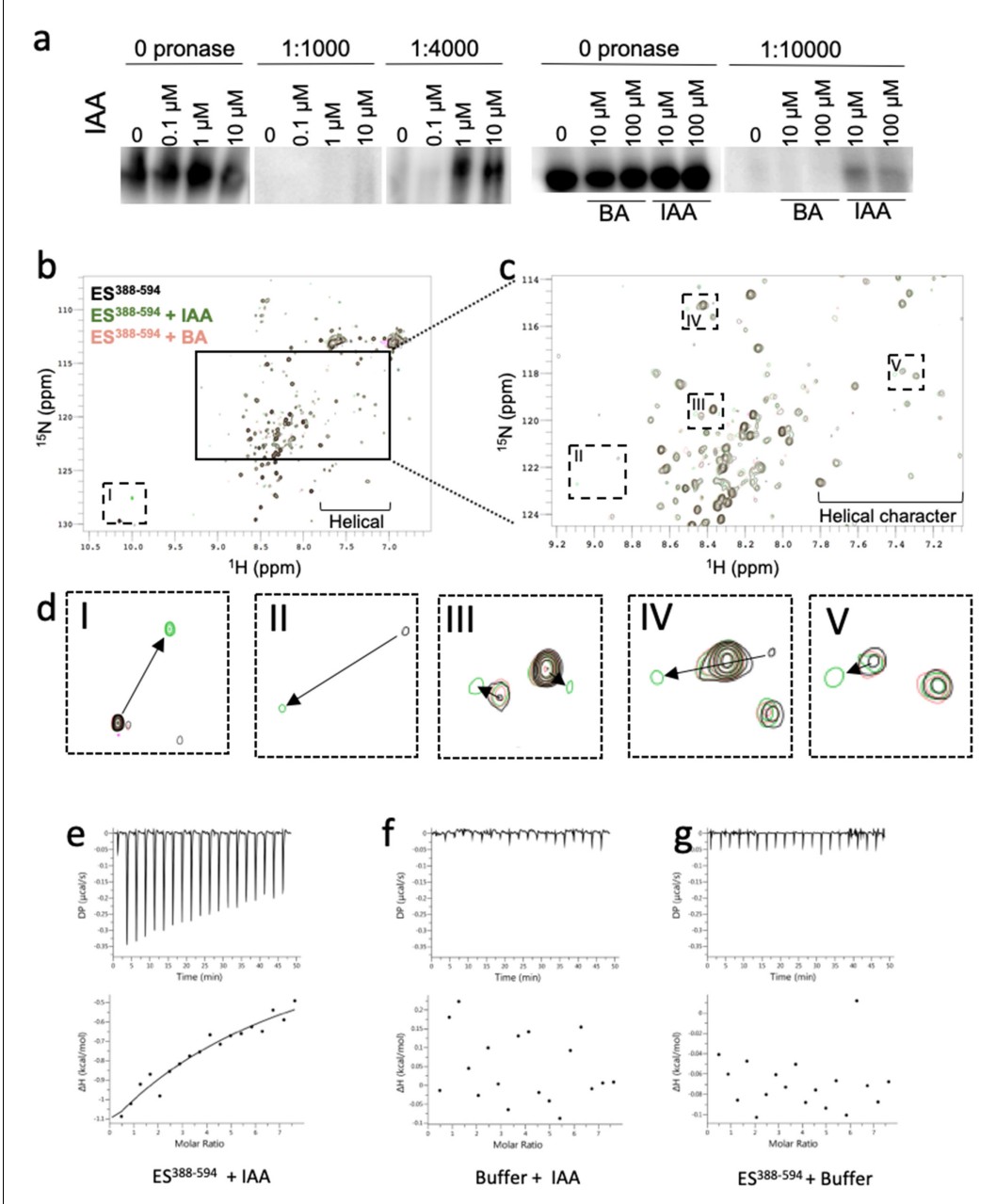

**Figure 1.** ETT directly binds auxin (IAA). (a) DARTS assay suggests that ETT binds IAA. Seedlings from an inducible two-component ETT-FLAG line were used for the protein isolation. Samples were treated with DMSO (mock), IAA and BA and digested by different concentrations of Pronase. Samples were further analysed by western blot with an anti-FLAG antibody. (b) HSQC-NMR performed with ES$^{388-594}$ protein either alone (black), with indole-3-acetic acid (IAA, green) or benzoic acid (BA, pink). ES$^{388-594}$: ligand, 1:10, 50 µM:500 µM. (c) Zoom-in of the indicated rectangular region in a. (d) Zoom-in of the specific shifts (labelled I-V) in the indicated dotted rectangles in b and c. Changes in chemical shifts are indicated by arrows from control to IAA treatment. (e–g) ITC spectre showing heat exchange between ES$^{388-594}$ protein and IAA (e), but not in controls (f, g). See *Figure 1— figure supplement 1* for parameters used in the HSQC-NMR experiment.

The online version of this article includes the following source data and figure supplement(s) for figure 1:

**Source data 1.** Parameters for HSQC NMR.
**Figure supplement 1.** Original western blot images for DARTS assays.
**Figure supplement 2.** Parameters for HSQC NMR experiment.

Previously, *PINOID* (*PID*) (*Benjamins et al., 2001*) and *HECATE1* (*HEC1*) (*Gremski et al., 2007*) were identified as ETT target genes (*Simonini et al., 2016*; *Simonini et al., 2017*) and both genes are upregulated in gynoecium tissue from the *ett-3* mutant compared to wild type (*Figure 2—figure supplement 1*). We also observed that expression of both genes is induced by IAA, but did not observe any additional induction beyond the constitutive upregulation in the *ett-3* mutant background (*Figure 2—figure supplement 1*). This ETT-dependent regulation does not require a functional TIR1/AFB machinery, since IAA-induction of *PID* and *HEC1* is still observed in *tir1/afb* mutant combinations, whereas the known TIR1/AFB-mediated auxin induction of the *IAA19* gene is completely abolished in these mutants (*Figure 2a–c*).

To further assess the TIR1/AFB independence of the ETT-mediated auxin signalling pathway, we exploited a recently-developed synthetic auxin-TIR1 pair (*Uchida et al., 2018*). In this system, the auxin-binding pocket of TIR1 has been engineered (ccvTIR1) to accommodate an IAA derivative bearing a bulky side chain (cvxIAA). By expressing the ccvTIR1 in a *tir1 afb2* mutant background, the canonical pathway will only respond to the addition of cvxIAA and not IAA (*Uchida et al., 2018*). We performed an expression analysis on *ccvTIR1* gynoecia treated ±cvxIAA and ±IAA as well as control

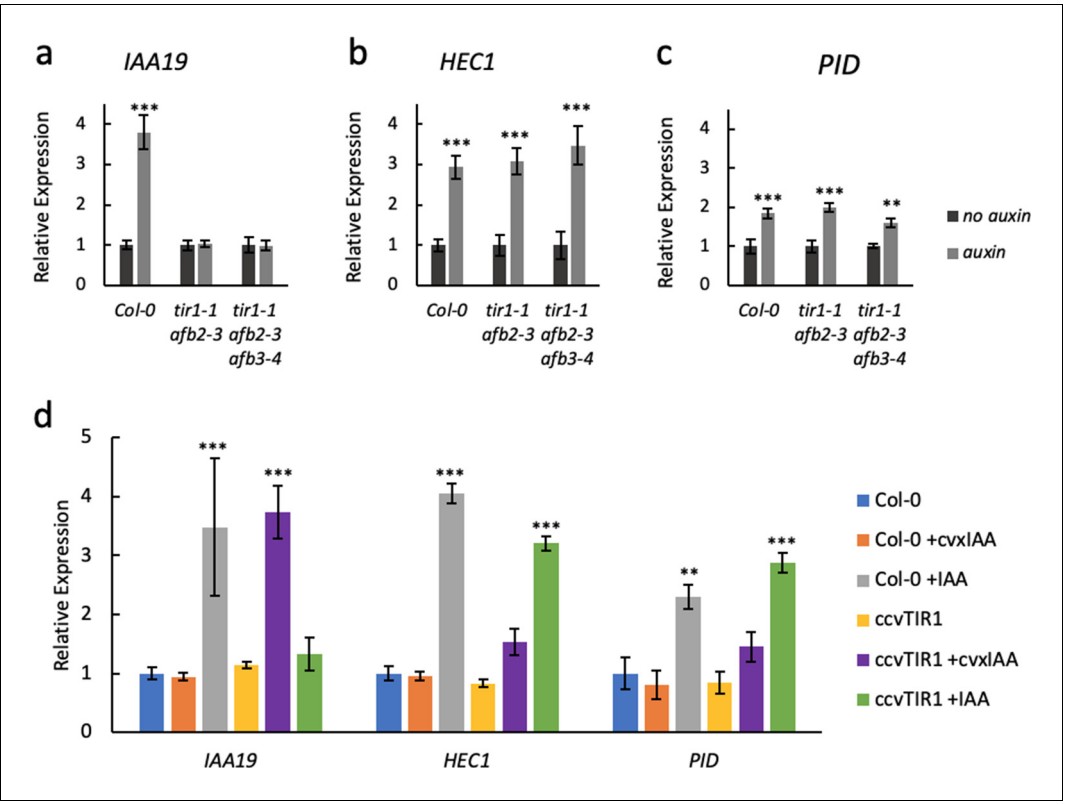

**Figure 2.** ETT regulates target gene expression independently of TIR1/AFB auxin receptors. Expression of the canonical auxin responsive *IAA19* gene (**a**) and the ETT-target genes *HEC1* (**b**) and *PID* (**c**) in control-treated (dark grey) or 100 µM IAA-treated (light grey) gynoecia assayed using qRT-PCR. (**a**) *IAA19* expression is up-regulated in response to auxin in wild-type gynoecia (*Col-0*) but not in *tir1/afb* double and triple mutants. The ETT-target genes *HEC1* and *PID* are up-regulated in response to auxin in both wild-type and auxin receptor mutants (**b c**). This suggests a TIR1/AFB independent regulation of these genes. (**d**) Expression of *IAA19*, *HEC1* and *PID* in response to treatment with 100 µM IAA and 100 µM cvxIAA in wild-type (*Col-0*) and *pTIR1:ccvTIR1* gynoecia in the *tir1 afb2* double mutant (*ccvTIR1*). The data confirm TIR1/AFB independent regulation of *HEC1* and *PID* in the gynoecium. ***p<0.0001; Shown are mean ± standard deviation of three biological replicates. See *Figure 2—source data 2* for statistical analyses.

The online version of this article includes the following source data and figure supplement(s) for figure 2:

**Source data 1.** Oligonucleotides used in this study.
**Source data 2.** Output of statistical tests.
**Figure supplement 1.** Expression of *HEC1* and *PID* in Col-0 and *ett-3*.

plants with the same treatments. In this experiment, *IAA19* served as a control gene whose expression is regulated in a TIR1/AFB-dependent manner (*Benjamins et al., 2001*). Indeed, *IAA19* was strongly upregulated by cvxIAA in the ccvTIR1 line, but not by IAA (*Figure 2d*). In contrast, *PID* and *HEC1* expression was not significantly affected by cvxIAA, whilst still responding to IAA in the ccvTIR1 background (*Figure 2d*). These data demonstrate that ETT-mediated auxin signalling can occur independently of the canonical TIR1/AFB signalling pathway.

In a phylogenetic analysis of ETT protein sequences across the angiosperm phylum, we identified a number of regions that are highly conserved (*Figure 3—figure supplement 1*). Unsurprisingly, the DNA-binding domain characteristic to B3-type TFs such as ARF proteins was conserved across all ETT proteins. Towards the C terminus of the ES domain we identified an EAR-like motif with a particularly high level of conservation (*Figure 3a*, *Figure 3—figure supplement 1a*). Ethylene-responsive element binding factor-associated Amphiphilic Repression (EAR) motifs are also found in Aux/IAA proteins. Interactions between Aux/IAA and members of the TOPLESS and TOPLESS-RELATED (TPL/TPR) family of co-repressors occur via this motif (*Szemenyei et al., 2008*). TPL/TPRs mediate their repressive effect by attracting histone deacetylases (HDACs) to promote chromatin condensation (*Krogan et al., 2012*). It was recently shown that ETT is required to keep Histone H3 at a deacetylated state at the gene locus of the meristem identity gene *SHOOT MERISTEMLESS* (*STM*) thereby repressing *STM* expression (*Chung et al., 2019*). Since ETT functions independently of the canonical

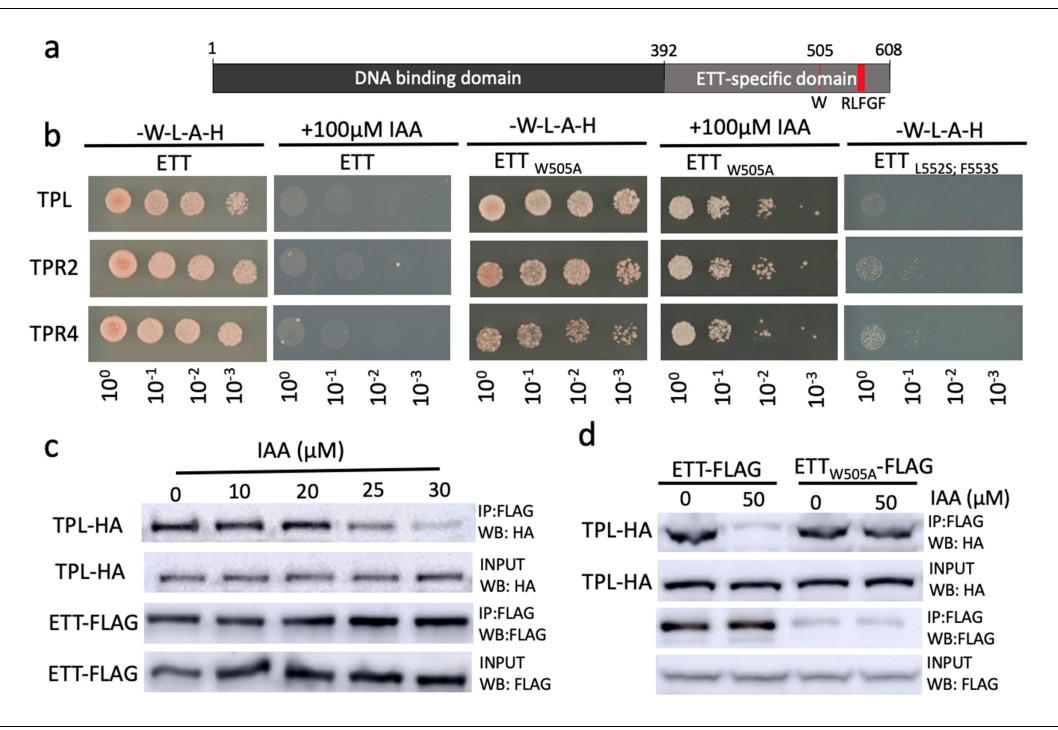

**Figure 3.** ETT interacts with members of the TPL/TPR co-repressor family in an auxin-sensitive manner. (a) Schematic representation of full-length ETT protein highlighting an EAR-like motif and tryptophan in position 505 (W) in the C-terminal ETT-specific domain. DBD, DNA-binding domain. (b) Y2H showing that ETT interacts with TPL, TPR2 and TPR4. These interactions depend on the identified C-terminal RLFGF motif and are auxin-sensitive. DBD, DNA-binding domain. (c) Co-IP revealing that ETT interacts with TPL in an auxin-sensitive manner with increasing IAA concentrations weakening the interaction. (d) Co-IP showing that mutating the tryptophan in position 505 of ETT (ETT$_{W505A}$) leads to loss of auxin sensitivity in the interaction between ETT and TPL. The online version of this article includes the following source data and figure supplement(s) for figure 3:

**Source data 1.** Output of statistical tests.
**Figure supplement 1.** ETT can interact with several members of the TPL/TPR co-repressor family through a conserved EAR-like motif.
**Figure supplement 2.** Interaction between ETT and TPL, TPR2 and TPR4 is auxin-sensitive and specific to IAA.
**Figure supplement 3.** Original western blot images for Co-immunoprecipitation assays.

auxin pathway, it is possible that its role in chromatin remodelling occurs via direct interaction with TPL/TPRs through the EAR-like motif. To test this, we carried out Yeast 2-Hybrid (Y2H) assays in which ETT was found to interact with TPL, TPR2 and TPR4 (*Figure 3b*, *Figure 3—figure supplement 1b*). Moreover, mutating residues in the EAR-like motif abolished the interactions demonstrating its requirement for the ETT-TPL/TPR interaction (*Figure 3b*).

Given that several ETT-protein interactions are affected by IAA and that part of the ETT transcriptome changes in response to IAA (*Simonini et al., 2016*; *Simonini et al., 2017*), we tested the IAA sensitivity of ETT-TPL/TPR interactions. In both Y2H and in co-immunoprecipitation (Co-IP) experiments, we observed that the interactions were reduced with increasing IAA concentrations (*Figure 3b,c* and *Figure 3—figure supplement 2*). Moreover, as described previously for other ETT-protein interactions, the sensitivity was specific to IAA as other auxinic compounds tested did not show this effect (*Figure 3—figure supplement 2*). Henceforth, 'auxin' will refer to IAA unless stated otherwise. These data suggest that in conditions with low auxin levels, ETT can interact with TPL/TPR proteins to repress the expression of target genes. An increase in cellular auxin causes ETT to bind auxin thereby undergoing a conformational change that abolishes interaction with TPL/TPR co-repressors.

The HSQC-NMR experiment described above indicate that the tryptophan in position 505 (W505) affects the direct interaction between the ETT protein and auxin. To test the role of W505 in the auxin-sensitive interaction with TPL/TPRs, we constructed a mutated version of ETT (W505A) and assessed the effect in protein interaction assays. Both Y2H and Co-IP experiments revealed that W505 is required for the ETT-TPL/TPR interaction to be sensitive to auxin (*Figure 3c,d*). The key importance of W505 is furthermore supported by a phylogenetic analysis of ETT protein sequences revealing that this residue is highly conserved in ETT proteins across angiosperms (*Figure 3—figure supplement 1a*).

TPL was originally identified as a key factor involved in setting up the apical-basal growth axis during embryo development (*Long et al., 2006*; *Smith and Long, 2010*). Large-scale interaction studies suggest that the five Arabidopsis TPL/TPRs have roles throughout plant development (*Krogan et al., 2012*; *Causier et al., 2012*). Whilst ETT has been implicated in a wide array of developmental processes (*Garcia et al., 2006*; *Marin et al., 2010*; *Kelley et al., 2012*; *Pekker et al., 2005*), the most dramatic phenotypes of *ett* loss-of-function mutants are observed during gynoecium development (*Sessions et al., 1997*; *Sessions and Zambryski, 1995*; *Nemhauser et al., 2000*). In accordance with this, *ETT* is highly expressed in the gynoecium (*Figure 4a*; *Simonini et al., 2016*). We produced reporter lines of *TPL*, *TPR2* and *TPR4* promoters fused to the *GUS* gene to test if they overlap with *ETT* expression in the gynoecium. Both *pTPL:GUS* and *pTPR2:GUS* exhibited strong expression in the apical part of the gynoecium where *ETT* is also expressed, while no *pTPR4: GUS* expression was observed (*Figure 4a–d*). Single loss-of-function mutants in *TPL* and *TPR2* do not show any abnormal phenotypes during gynoecium development. However, the *tpl tpr2* double mutant has defects in the development of the apical gynoecium similar to *ett* mutants (*Figure 4e–g*) demonstrating that *TPL* and *TPR2* function redundantly in gynoecium development. Together with the protein interaction data and the overlapping expression patterns, these results suggest that ETT and TPL/TPR2 cooperate to regulate gynoecium development.

TPL was shown previously to recruit histone deacetylase, HDA19, during early Arabidopsis flower development to keep chromatin in a repressed state (*Krogan et al., 2012*). Moreover, HDA19 was also recently shown to participate in the repression of *STM* (*Chung et al., 2019*). In that study, ETT was shown to recruit HDA19 to the *STM* promoter, although not via direct protein interaction. Here, our analysis of gynoecia from the *hda19-4* mutant demonstrate that HDA19 is also required for gynoecium development as the *hda19-4* mutant has strong style defects (*Figure 4h*). In agreement with this, the *HDA19* gene was highly expressed in gynoecium tissue, whereas another member of the *HDA* gene family, *HDA6*, was not (*Figure 4—figure supplement 1*). Moreover, HDA19 recruitment likely involves ETT, since expression of the ETT target genes, *PID* and *HEC1*, is increased in the *tpl tpr2* and *hda19-4* mutants compared to wild type. Similar to the *ett* mutant, auxin treatments failed to further induce expression in these mutants (*Figure 4i,j*). These observations suggest that ETT, TPL/TPR2 and HDA19 function in conjunction to control gene expression during gynoecium development.

To test the direct interaction of ETT, TPL and HDA19 on chromatin, we performed Chromatin-Immunoprecipitation (ChIP) using reporter lines expressing GFP fusion protein. Although only ETT is

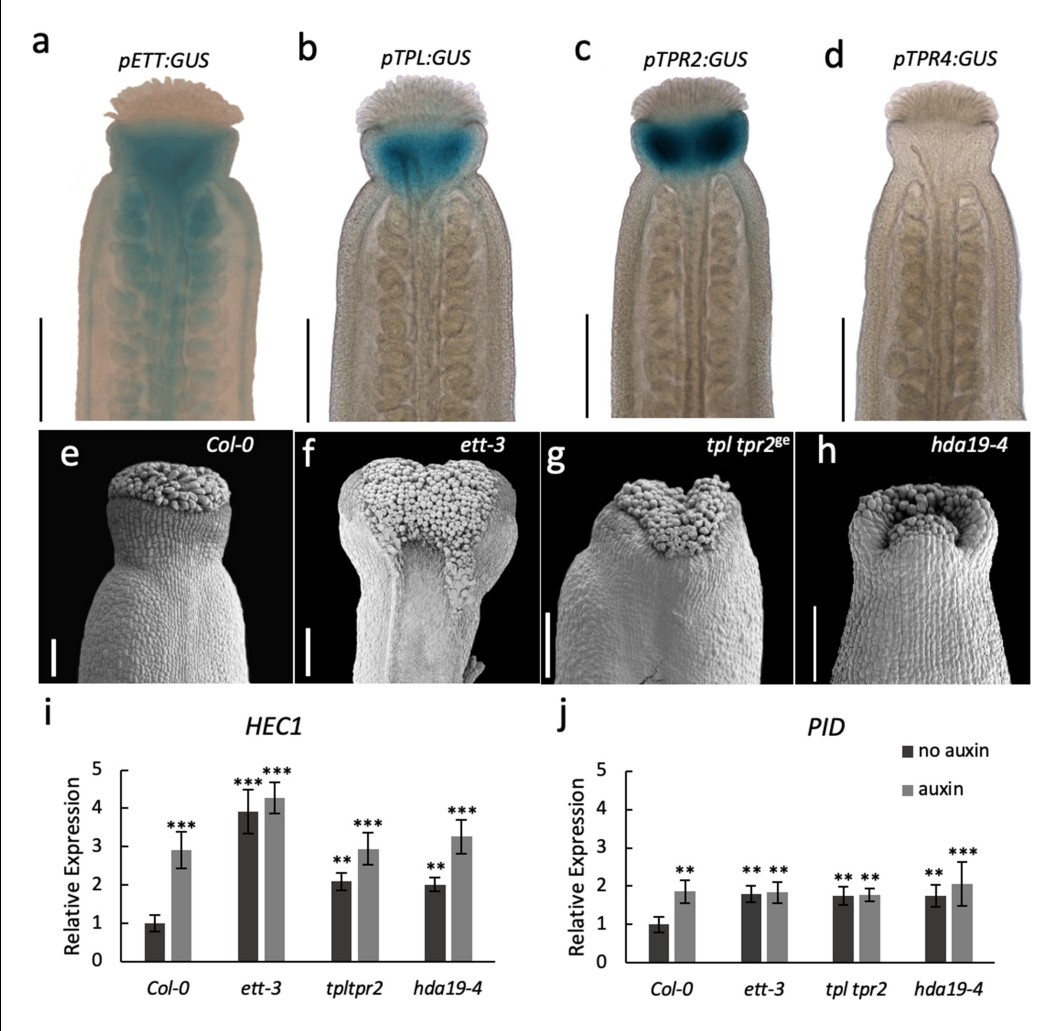

**Figure 4.** ETT, TPL/TPR2 and HDA19 co-operatively regulate gene expression to facilitate gynoecium development. (a–d) Promoter GUS expression analysis of *pETT:GUS* (a), *pTPL:GUS* (b), *pTPR2:GUS* (c) and *pTPR4: GUS* (d) revealed that *ETT*, *TPL* and *TPR2* but not *TPR4* are co-expressed in the Arabidopsis style. Scale bar = 300 μm. (e–h) Gynoecium phenotypes of wild-type (e), *ett-3* (f) *tpl tpr2ge* (g) and *hda19-4* (h). Scale bar = 100 μm. (i j,) *HEC1* (i) and *PID* (j) are constitutively mis-regulated in *ett-3*, *tpl tpr2ge* and *hda19-4* gynoecia. This misregulation is unaffected by treatment with 100 μM IAA. ***p-values<0.0001; Shown are mean ± standard deviation of three biological replicates. Differences between untreated and IAA-treated mutants are not significant. See *Figure 4— source data 1* for statistical analyses.

The online version of this article includes the following source data and figure supplement(s) for figure 4:

**Source data 1.** Output of statistical tests.

**Figure supplement 1.** Expression of *TPL*, *TPRs* and *HDAs* genes in the gynoecium.

expected to bind DNA, ChIP followed by qPCR revealed that all three proteins associate with DNA elements in the same regions of the promoters of *PID* and *HEC1* (*Figure 5a–c*). Moreover, while ETT association with these promoter regions was largely unaffected in the presence of IAA, TPL and HDA19 interactions are reduced upon auxin treatment (*Figure 5a–c*). This supports a model in which ETT recruits TPL/TPR2 and HDA19 to ETT target loci to keep chromatin in a condensed state through histone deacetylation. When auxin levels increase, the ETT-TPL/TPR2 interaction is broken and TPL/TPR2 and HDA19 dissociate from the loci, presumably preventing HDA19 from deacetylating histones. To test this, we assayed for H3K27 acetylation, which is a substrate for HDA19. H3K27 acetylation increased in the absence of ETT and upon treatment with auxin. This occurred in the same regions of the *PID* and *HEC1* promoters where the proteins were found to associate

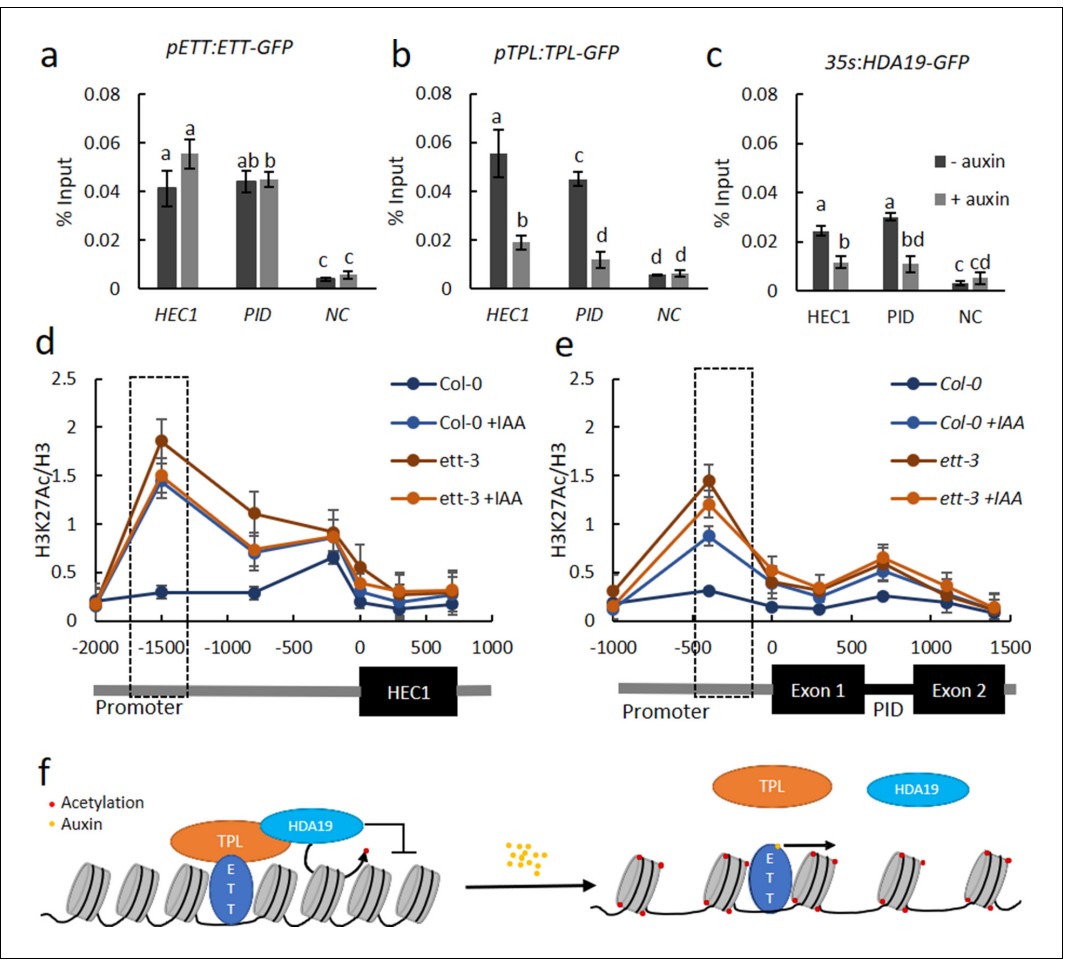

**Figure 5.** ETT, TPL and HDA19 co-operatively regulate *HEC1* and *PID* by modulating chromatin acetylation. (a–c) Chromatin immunoprecipitation (ChIP) shows ETT (a), TPL (b) and HDA19 (c) binding to conserved regions of *HEC1* and *PID* loci in the absence of IAA (dark grey columns). In the presence of IAA (light grey columns), ETT association is unchanged, while both TPL and HDA19 interactions are reduced. A *WUS* gene fragment served as negative control (NC). (d e,) H3K27ac accumulation (from ChIP analysis) along the *HEC1* (d) and *PID* (e) loci in wild-type (*Col-0*) and *ett-3* plants ± treatment with 100 μM IAA. Numbers on the x axes are distances to the Transcription Start Site (TSS). The schematic of the loci is shown below each panel. Dashed boxes represent ETT binding regions. (f) Qualitative schematic illustration of alternative TIR1/AFB independent auxin signaling. Under low auxin conditions an ETT-TPL-HDA19 complex binds to ETT-target genes keeping their chromatin environments repressed, through de-acetylation. High nuclear auxin concentrations abolish the ETT-TPL-HDA19 complex through direct ETT-auxin interaction. This leads to an accumulation of histone acetylation and up-regulation of ETT-target genes. Values in (a–e) are means ± standard deviation of three biological replicates. See *Figure 5—source data 1* for statistical analyses.

The online version of this article includes the following source data for figure 5:

**Source data 1.** Output of statistical tests.

(*Figure 5d,e*). In agreement with ETT mediating the association of TPL/TPR and HDA19 with these regions, there was no further increase of acetylation in the *ett-3* mutant upon treatment with auxin (*Figure 5d,e*).

## Discussion

The data presented in this paper provide molecular insight into how auxin levels are translated into changes in gene expression of ETT target genes. Our data lead to a model in which low levels of auxin maintain ETT associations with TPL/TPR2 to repress gene expression via H3K27 deacetylation.

As auxin levels increase, TPL/TPR2 (and hence HDA19) disassociate from ETT, promoting H3K27 acetylation (*Figure 5f*). This model molecularly underpins the published association between auxin dynamics and *PID* expression at the gynoecium apex where *PID* is repressed at early stages of development to allow symmetry transition, but subsequently de-repressed as auxin levels rise to facilitate polar auxin transport (*Simonini et al., 2016*; *Moubayidin and Ostergaard, 2014*). Whilst this model explains the data on ETT-mediated auxin signalling during gynoecium development, it is possible that the effect of auxin on ETT varies depending on the developmental context. Moreover, the concentrations of auxin required to mediate its effect on ETT is likely to differ between diverse ETT-containing complexes.

The direct binding of auxin allows ETT to switch the chromatin locally between repressive and de-repressive states. The recent report of ETT negatively regulating *STM* expression (*Chung et al., 2019*) provides molecular evidence that ETT can function as a repressor. Whether ETT also has a role in recruiting activating components is yet to be clarified. Indeed, different modes of actions are possible that could depend on other interacting partners or auxin levels and even be target-gene specific. Nevertheless, the effect of auxin on ETT-mediated regulation of the genes studied here is instantly reversible, making it possible to switch between states, immediately responding to changes in auxin levels. This feature, which is reminiscent of animal hormonal signalling pathways such as the Thyroid Hormone and Wnt/ß-catenin pathways (*Tsai and O'Malley, 1994*; *Gammons and Bienz, 2018*), may be particularly important in controlling changes in tissue polarity during plant organogenesis as observed in the Arabidopsis gynoecium (*Moubayidin and Ostergaard, 2014*).

The identification of a direct auxin-ETT interaction to control gene expression adds an additional layer of complexity to auxin action, which contributes towards explaining how auxin imparts its effect on highly diverse processes throughout plant development. In a broader context, this work also opens for the exciting possibility that direct transcription factor-ligand interactions is a general feature in the control of gene expression in plants as found in animals.

# Materials and methods

### Key resources table

| Reagent type (species) or resource | Designation | Source or reference | Identifiers | Additional information |
|---|---|---|---|---|
| Gene (*Arabidopsis thaliana*) | *ETTIN* (*ETT*) | The Arabidopsis Information Resource | AT2G33860 | |
| Gene (*Arabidopsis thaliana*) | *TOPLESS* (*TPL*) | The Arabidopsis Information Resource | AT1G15750 | |
| Gene (*Arabidopsis thaliana*) | *TOPLESS- RELATED 1 (TPR1)* | The Arabidopsis Information Resource | AT1G80490 | |
| Gene (*Arabidopsis thaliana*) | *TOPLESS-RELATED 2 (TPR2)* | The Arabidopsis Information Resource | AT3G16830 | |
| Gene (*Arabidopsis thaliana*) | *TOPLESS-RELATED 3 (TPR3)* | The Arabidopsis Information Resource | AT5G27030 | |
| Gene (*Arabidopsis thaliana*) | *TOPLESS-RELATED 4 (TPR4)* | The Arabidopsis Information Resource | AT3G15880 | |
| Gene (*Arabidopsis thaliana*) | *HISTONE DEACETYLASE 6 (HDA6)* | The Arabidopsis Information Resource | AT5G63110 | |
| Gene (*Arabidopsis thaliana*) | *HISTONE DEACETYLASE 19 (HDA19)* | The Arabidopsis Information Resource | AT4G38130 | |
| Gene (*Arabidopsis thaliana*) | *HECATE 1 (HEC1)* | The Arabidopsis Information Resource | AT5G67060 | |
| Gene (*Arabidopsis thaliana*) | *PINOID* (*PID*) | The Arabidopsis Information Resource | AT2G34650 | |
| Genetic reagent *Arabidopsis thaliana* | *Col-0* | widely distributed | | |
| Genetic reagent *Arabidopsis thaliana* | *hda19-4* | *Kim et al., 2008* | SALK_13944 | |

*Continued on next page*

*Continued*

| Reagent type (species) or resource | Designation | Source or reference | Identifiers | Additional information |
|---|---|---|---|---|
| Genetic reagent *Arabidopsis thaliana* | *tir1-1 afb2-3 afb3-4* | *Parry et al., 2009* | | |
| Genetic reagent *Arabidopsis thaliana* | *tpl* | European Arabidopsis Stock Centre | SALK_034518C | |
| Genetic reagent *Arabidopsis thaliana* | *ett-3* | *Simonini et al., 2017*; *Sessions et al., 1997* | AT2G33860 | |
| Genetic reagent *Arabidopsis thaliana* | *pETT:GUS* | *Ng et al., 2009* | AT2G33860 | |
| Genetic reagent *Arabidopsis thaliana* | *pETT:ETT-GFP* | *Simonini et al., 2016* | AT2G33860 | |
| Genetic reagent *Arabidopsis thaliana* | *pTPL:TPL-GFP* | *Pi et al., 2015* | AT1G15750 | |
| Genetic reagent *Arabidopsis thaliana* | *p35S:HDA19-GFP* | *Pi et al., 2015* | AT4G38130 | |
| Genetic reagent *Arabidopsis thaliana* | *pTIR1:ccvTIR1* | *Szemenyei et al., 2008* | | |
| Genetic reagent *Arabidopsis thaliana* | *tpl tpr2*[ge] | this paper | AT1G15750, AT3G16830 | Further details in the Materials and methods section |
| Genetic reagent *Arabidopsis thaliana* | *pTPL:GUS* | this paper | AT1G15750 | Further details in the Materials and methods section |
| Genetic reagent *Arabidopsis thaliana* | *pTPR2:GUS* | this paper | AT3G16830 | Further details in the Materials and methods section |
| Genetic reagent *Arabidopsis thaliana* | *pTPR4:GUS* | this paper | AT3G15880 | Further details in the Materials and methods section |
| Genetic reagent *Arabidopsis thaliana* | *p35S:CDGVG p6xGAL4UAS:ETT-FLAG* | this paper | | Further details in the Materials and methods section |
| Antibody | Mouse anti-FLAG | Abcam | ab49763 | monoclonal, conjugated with HRP |
| Antibody | Mouse anti-HA | Abcam | ab173826 | monoclonal, conjugated with HRP |
| Antibody | Mouse anti-GFP | Roche | 11814460001 | monoclonal |
| Antibody | Rabbit anti-H3K27ac | Abcam | ab4729 | polyclonal |
| Antibody | Rabbit anti-H3 | Abcam | ab1791 | polyclonal |
| Recombinant DNA reagent | pGWB14; *p35s:HA-TPL* | *Espinosa-Ruiz et al., 2017* | | |
| Recombinant DNA reagent | pICH47732; *p35s: ETT-3xFLAG* | this paper | | Backbone Addgene #48000 |
| Recombinant DNA reagent | pICH47732; *p35s: ETT_{W505A}-3xFLAG* | this paper | | Backbone Addgene #48000 |
| Recombinant DNA reagent | pGADT7; *ETT* | this paper | | Further details in the Materials and methods section |
| Recombinant DNA reagent | pGADT7; *ETT_{W505A}* | this paper | | Further details in the Materials and methods section |
| Recombinant DNA reagent | pGADT7; *ETT_{L552S; F553S}* | this paper | | Further details in the Materials and methods section |

*Continued on next page*

*Continued*

| Reagent type (species) or resource | Designation | Source or reference | Identifiers | Additional information |
|---|---|---|---|---|
| Recombinant DNA reagent | pGBKT7; *TPL* | *Causier et al., 2012* | | |
| Recombinant DNA reagent | pGBKT7; *TPR1* | *Causier et al., 2012* | | |
| Recombinant DNA reagent | pGBKT7; *TPR2* | *Causier et al., 2012* | | |
| Recombinant DNA reagent | pGBKT7; *TPR3* | *Causier et al., 2012* | | |
| Recombinant DNA reagent | pGBKT7; *TPR4* | *Causier et al., 2012* | | |
| Recombinant DNA reagent | pESPRIT; *ES$^{388-594}$* | *Simonini et al., 2018* | | |

## Plant materials and treatments

Plants were grown in soil at 22˚C in long day conditions (16 hrs day/8 hr dark). All mutations were in the *Col-0* background. Mutant alleles described before include e*tt-3* (*Sessions et al., 1997*; *Simonini et al., 2016*), *hda19-4* (SALK_139443) (*Kim et al., 2008*), *pETT:GUS* (*Ng et al., 2009*), *pETT:ETT-GFP* in *ett-3* (*Simonini et al., 2016*), *pTPL:TPL:GFP* (*Pi et al., 2015*), *p35S:HDA19:GFP* (*Pi et al., 2015*), *pTIR1:ccvTIR1* in *tir1-1 afb2-3* (*Szemenyei et al., 2008*) and *tir1-1 afb2-3 afb3-4* (*Parry et al., 2009*). The *tpl* mutant (SALK_034518C) was obtained from the European Arabidopsis Stock Centre.

For both expression and ChIP analysis, auxin treatments were applied by spraying bolting *Col-0* and *ett-3* inflorescences with a solution containing 100 µM IAA (Sigma) or cvxIAA and 0.015% Silwet L-77 (De Sangosse Ltd.). Treated samples were returned to the growth room and incubated for two hours.

## Expression analysis

Quantitative Real time PCR (qRT-PCR) was used for expression analysis. RNA was extracted from floral buds using the RNeasy mini kit (Qiagen). Using the SuperScript IV First-Strand Synthesis kit (ThermoFisher), cDNA was synthesised from 1 µg of total RNA. Subsequently, qRT-PCR was carried out using SYBR Green JumpStart Taq ReadyMix (Sigma) using the appropriate primers (*Figure 2—source data 1*). Relative expression values were determined using the $2^{-\Delta\Delta Ct}$ method (*Livak and Schmittgen, 2001*). Data were normalised to *POLYUBIQUITIN 10* (*UBQ10*/AT4G05320) expression.

## ETT protein analysis by alignment

Published *ETT* sequences of 22 Angiosperm species were retrieved from Phytozome version 12 (*Goodstein et al., 2012*). Nucleotide sequences were translated and aligned using MUSCLE in Geneious version 6.1.8 (*Kearse et al., 2012*). The EAR-like motif was extracted as a sequence logo (*Figure 3a*; *Figure 3—figure supplement 1*).

## Generation of the *tpl tpr2* CRISPR mutant

The *tpl tpr2$^{ge}$* mutant was generated using CRISPR/Cas9 technology by a method previously described (*Castel et al., 2019*). Briefly, for the construction of the RNA-guided genome-editing plasmid, DNA sequences encoding the gRNA adjacent to the PAM sequences were designed to target two specific sites in *TPR2* (AT3G16830). DNA-oligonucleotides (*Figure 2—source data 1*) containing the specific gRNA sequence were synthesised and used to amplify the full gRNA from a template plasmid (AddGene #46966). Using Golden Gate cloning (*Engler et al., 2014*) each gRNA was then recombined in a L1 vector downstream of U6 promoter (*Goodstein et al., 2012*). Finally, the resulting gRNA plasmids were then recombined with a L1 construct containing *pYAO:Cas9_3:E9t* (*Castel et al., 2019*) (kindly provided by Jonathan Jones) and a L1 construct containing Fast-Red selection marker (AddGene #117499) into a L2 binary vector (AddGene #112207).

The construct was transformed into *Agrobacterium tumefaciens* strain GV3101 by electroporation, followed by plant transformation by floral dip into the *tpl* single mutant (**Clough and Bent, 1998**). Transgenic T1 seeds appear red under UV light and were selected under a Leica M205FA stereo microscope. T1 plants were genotyped using PCR and the *TPR2* locus sequenced (Oligonucleotides in *Figure 2—source data 1*). Genome-edited plants were selected and the next generation grown (T2). Seeds of this generation were segregating in a 3:1 ratio for the transgene. Transgene negative plants were selected and grown on soil. To identify homozygous mutations T2 plants were genotyped. The T3 generation was again checked for the absence of the transgene.

## Protein interaction

For Yeast-two-Hybrid (Y2H) assays coding sequences were cloned into pDONR207 and recombined into the pGDAT7 and pGBKT7 (Clontech). Using the co-transformation techniques (**Engler et al., 2014**) these constructs were transformed into the AH109 strain (Clontech). Transformations were selected on Yeast Selection Medium (YSD) lacking Tryptophan (W) and Leucine (L) at 28°C for 3–4 days. Transformed yeast cells were serially diluted ($10^0$, $10^{-1}$, $10^{-2}$ and $10^{-3}$) and dotted on YSD medium lacking Tryptophan (W), Leucin (L), adenine (A) and Histidine (H) to test for interaction. To examine interaction strength 3-amino-1,2,4-triazole (3-AT) was supplemented to the YSD (-W-L-A-H) medium with different concentrations (0, 5, 10 mM). To determine the effect of auxinic compounds on the protein-protein interactions benzoic acid (BA), IAA, NAA and 2,4D (all Sigma) were dissolved in ethanol and added directly to the medium at the desired concentrations. Pictures were taken after 3 days of growth at 28°C.

For the β-Galactosidase assay, transgenic yeast was grown in liquid YSD (-W-L) medium supplemented with /-out 100 µM IAA or NAA, to an $OD_{600}$ of 0.5. The cells were then harvested and lysed using 150 µL Buffer Z with β-mercaptoethanol (100 mM Phosphate buffer pH 7, 10 mM KCl, 1 mM $Mg_2SO_4$, β-mercaptoethanol 50 mM), 50 µL chloroform and 20 µL of 0.1% SDS. After lysis, the sample was incubated with 700 µL pre-warmed ONPG solution (1 mg/mL ONPG (o-Nitrophenyl-β-D-Galactopyranoside, Sigma) prepared in Buffer Z without β-mercaptoethanol at 37°C until a yellow colour developed in the samples without auxin treatment. After stopping all reactions (using 500 µL $Na_2CO_3$) the supernatant was collected and $OD_{405}$ determined. The β-Galactosidase activity was calculated as follows: (A405*1000)/(A600*min*mL).

For co-immunoprecipitation, ETT-FLAG was generated using Golden Gate cloning (**Kearse et al., 2012**) by recombining a previously described L0 clone for *ETT* (**Simonini et al., 2016**) with a 35S promoter (AddGene #50266), a C-terminal 3xFLAG epitope (AddGene #50308) and a Nos-terminator (AddGene #50266) into a L1 vector (AddGene #48000). The pGWB14 TPL-HA construct was provided by Salomé Prat and has been used in previous studies (**Espinosa-Ruiz et al., 2017**). The epitope-tagged proteins were transiently expressed in four-week-old *N. benthamiana* leaves for two days. Co-immunoprecipitation was performed as described previously (**Egea-Cortines et al., 1999**). After harvest, 1 g of fresh leaf tissue was ground in liquid nitrogen. The powder was homogenised for 30 min in two volumes of extraction buffer (10% glycerol, 25 mM Tris-HCl pH 7.5, 1 mM EDTA, 150 mM NaCl, 0.15% NP-40, 1 mM PMSF, 10 mM DTT, 2% Polyvinylporrolidone, 1x cOmplete Mini tablets EDTA-free Protease Inhibitor Cocktail (Roche). The homogenised samples were cleared by centrifugation at 14,000 x g for 10 min and cleared lysates were incubated for 2 hr with 20 µl anti-FLAG M2 magnetic beads (SIGMA-ALDRICH, M8823; lot: SLB2419). The beads were washed five times with IP buffer (10% glycerol, 25 mM Tris-HCl pH 7.5, 1 mM EDTA, 150 mM NaCl, 0.15% NP-40, 1 mM PMSF, 1 mM DTT, 1x cOmplete Mini tablets EDTA-free Protease Inhibitor Cocktail (Roche)) and proteins were eluted by adding 80 µl 2x SDS loading buffer followed by an incubation at 95°C for 10 min. To examine auxin sensitivity, 4 g of fresh leaf tissue was collected, ground in liquid nitrogen and protein was extracted. The lysate was then divided according to the number of treatments. The desired concentration of IAA or NAA was added to each of the cleared lysates before the anti-FLAG M2 magnetic beads were added. IAA or NAA at the desired concentration was also supplemented to the IP buffer during the washes. The eluates were analysed by western blot using an anti-FLAG antibody (M2, Abcam, ab49763, Lot: GR3207401-3) or an anti-HA antibody (Abcam, ab173826, Lot: GR3255539-1). Both antibodies were used as 1:10000 dilutions. The antibodies were validated by the manufacturer.

## Scanning electron microscopy

Whole inflorescences of *Col-0*, *ett-3*, *tpl tpr2^{ge}* and *hda19-4* were fixed overnight in FAA (3.7% form-aldehyde, 5% glacial acetic acid, 50% ethanol) and dehydrated through an ethanol series (70% to 100%) as described previously (*Moubayidin and Ostergaard, 2014*). The samples were then critical point-dried, gynoecia dissected and mounted. After gold coating, samples were examined with a Zeiss Supra 55VP Field Emission Scanning electron microscope using an acceleration voltage of 3 kV.

## *TPL*, *TPR2* and *TPR4* reporter lines

For the construction of the promoter:GUS reporter plasmids of *TPL*, *TPR2* and *TPR4*, 2.5 kb of promoter sequences were isolated from genomic DNA and inserted upstream of the ß-glucoronidase gene of pCambia1301 vectors using the In-Fusion Cloning Recombinase kit (Clontech). The constructs were transformed into *Agrobacterium tumefaciens* strain GV3101 by electroporation, followed by plant transformation by floral dip into *Col-0* (*Clough and Bent, 1998*).

The GUS histochemical assay was performed in at least three individual lines per construct. Inflorescences of each GUS line were pre-treated with ice cold acetone for 1 hr at −20°C and washed two times for 5 min with 100 mM sodium phosphate buffer followed by one wash with sodium phosphate buffer containing 1 mM $K_3Fe(CN)_6$ and 1 mM $K_4Fe(CN)_6$ (both Sigma) at room temperature. Subsequently, samples were vacuum infiltrated for 5 min with X-Gluc solution (100 mM sodium phosphate buffer, 10 mM EDTA, 0.5 mM $K_3Fe(CN)_6$, 3 mM $K_4Fe(CN)_6$, 0.1% Triton X100) containing 1 mg/ml of ß-glucoronidase substrate X-gluc (5-bromo-4-chloro-3-indolylglucuronide, Melford) and incubated at 37°C. *pTPL:GUS* samples were incubated for 20 min and *pTPR2:GUS* lines for 45 min to prevent overstaining. *pTPR4:GUS* lines were incubated for 16 hr. After staining, the samples were washed in 70% ethanol until chlorophyll was completely removed. Gynoecia were dissected and mounted in chloral hydrate (Sigma). Samples were analysed using a Leica DM6000 light microscope.

## Chromatin immunoprecipitation

Transcription factor ChIP was performed in biological triplicates using the *pETT:ETT:GFP*, *pTPL:TPL:GFP* and *p35S:HDA19-GFP* lines and data analysed as described previously (*Chung et al., 2019*). Additionally, a *WUS* gene fragment (*WUSp4* amplicon) was used as a negative control for ETT binding (*Liu et al., 2014*). IP was conducted using the anti-GFP antibody (Roche, 11814460001, Lot: 19958500) and Pierce Protein G magnetic beads (ThermoFisher, 88847, Lot: SI253639) were used for IP.

Histone acetylation ChIP was carried out and data were analysed as described previously (*Qüesta et al., 2016*). The experiment was carried out in triplicate using 3 g auxin-treated or untreated *Col-0* or *ett-3* inflorescent tissue. The antibodies used for IP were anti-H3K27ac antibodies (Abcam, ab4729, Lot: GR3231937-1) and anti-H3 (Abcam, ab1791, Lot: GR310541-1). All antibodies were validated by the manufacturers.

In all ChIP experiments, DNA enrichment was quantified using quantitative PCR (qPCR) with the appropriate primers (*Figure 2—source data 1*). In case of H3K27ac, H3 was used as an internal control and the data represented as ratio of (H3K27ac at *HEC1* or *PID* divided by H3 at *HEC1 or PID*).

## Statistical analyses and replication

In all graphs error bars represent the standard deviation of the mean for all numerical values. QRT-PCR and ChIP experiments have been carried out at least in triplicate. The data presented here show an average of three replicates. For qRT-PCR data were analysed using one-way ANOVA with post-hoc Tukey multiple comparison test. $ChIP_{qPCR}$ data were analysed using two-way ANOVA with post hoc Bonferroni multiple comparison test. All output of statistical tests can be found in the source data files. All statistical tests were carried out using GraphPad Prism Version 5.04 (La Jolla California USA, www.graphpad.com).

## Drug Affinity Responsive Target Stability (DARTS) assay

The DARTS assay to evaluate the stability of ETTIN-FLAG in the presence of IAA was performed as described previously (*Lomenick et al., 2011*; *Kania et al., 2018*; *Tan et al., 2020*) with plant material of a two-component inducible ETTIN-FLAG line. This ETT-FLAG line was generated using Golden

Gate cloning (*Engler et al., 2014*) by recombining a previously described L0 clone for *ETT* (*Simonini et al., 2016*) with a 6xGAL4UAS promoter, a C-terminal 3xFLAG epitope (AddGene #50308) and a Nos-terminator (AddGene #50266) into an L1 vector (AddGene #48000). This inducible ETTIN-FLAG cassette was combined with a dexamethasone inducer cassette (containing CD-GVG under a 35S promoter) and a phosphinothricin-resistance cassette in an L2 binary vector (AddGene #48015). The construct was transformed into *Agrobacterium tumefaciens* strain GV3101 by electroporation, followed by floral dip-mediated plant transformation into Arabidopsis *Col-0*. For the DARTS assay, seeds were surface sterilised using chlorine gas and germinated in 25 mL liquid ½ MS medium containing 1% sucrose in 100 mL Erlenmeyer flasks under gently shaking. ETTIN-FLAG expression was induced 7 days post germination by adding 10 µM dexamethasone to the growth medium and seedlings were harvested 48 hr post induction. Total plant material was ground in liquid nitrogen and resuspended in protein extraction buffer (25 mM Tris-HCl, pH 7.5; 150 mM NaCl; 0.1% IGEPAL CA-630 and Roche cOmplete protease inhibitor cocktail, EDTA-free) in a 1:1 (w/v) ratio, followed by a centrifuging step to discard the plant debris. Protein concentration was determined with Quick Start Bradford 1x reagent (Bio-Rad) and adjusted to 5 mg/mL by adding extraction buffer. The lysate was then split into different reaction tubes and incubated with the respective chemical (DMSO mock, IAA or BA) at the indicated concentrations for 30 min at room temperature with slow mixing. The treated aliquots were further aliquoted and mixed with different dilutions of Pronase (Roche), prepared in Pronase buffer (25 mM Tris-HCl, pH 7.5 and 150 mM NaCl) to achieve the aimed for ratio of total enzyme to total protein. After incubation for 30 min at room temperature, the proteolytic digestion was terminated by adding protease inhibitor cocktail (cOmplete, Roche) and the samples were kept on ice for 10 min. The protein samples were then mixed with 4x NuPAGE LDS sample buffer (Invitrogen) and heated at 80°C for 10 min. The protein samples were analysed by Western blot (TGX gels Bio-Rad, TurboTransfer Bio-Rad PVDF membranes), visualized using an anti-FLAG-HRP antibody (1:1000 – Sigma A8592). HRP activity was detected using the Supersignal Western Detection Reagents (Thermo Scientific) and imaged with a GE Healthcare Amersham 600 RGB system.

## Protein production

The ES domain, $ES^{388-594}$, protein was isotopically labelled in preparation for NMR analysis. The ES domain was expressed for as a fusion protein with a 6x Histidine tag in minimal media with $^{15}N$ ammonium chloride. The $^{15}N$ isotope labelling of the expressed protein involved a 125-fold dilution of cell culture in enriched growth media into minimal media with $^{15}N$ ammonium chloride and grown for 16 hr (37 °C / 200 rpm); followed by a further 40-fold dilution into minimal media for the final period of cell growth and protein expression (induced with L-arabinose 0.2 % w/v / 18 °C / 200 rpm and grown for a further 12 hr). The fusion protein was isolated from soluble cell lysate by Co-NTA affinity chromatography with two His-Trap 1 mL TALON Crude columns (GE Healthcare Life Sciences, 28953766). Chromatography buffers contained sodium phosphate 20 mM pH 8.0, NaCl 500 mM and either no-imidazole or 500 mM imidazole for wash and elution buffers respectively. The majority of the non-specifically bound protein was removed by passing 20 mL of the wash buffer through the columns. The protein eluted on a gradient of increasing imidazole concentration of up to 30% elution buffer over 20 mL.

For ITC, the $ES^{388-594}$ protein was grown in LB medium, cells harvested and lysed as described above. After centrifugation of the sonicated cell suspension, the pellet was resuspended and washed in Wash Buffer (50 mM Tris-HCl, pH 8.0; 0.1 mM EDTA; 5% (v/v) Glycerol; 0.1 mM DTT; 50 mM NaCl; 2% (w/v) NaDOC) for 1 hour at 4°C followed by centrifugation (11,000 g for 20 min at 4 °C). The resulting pellet was again washed in Wash Buffer and centrifuged as described previously. The pellet was resuspended in 20mL of Solubilisation Buffer (50 mM Tris-HCl, pH 8.0; 0.1 mM EDTA; 5% (v/v) Glycerol; 0.1 mM DTT; 50 mM NaCl; 0.25% Sarkosyl (N-Lauroyl sarcosine) and stirred at 4°C for 1h. Subsequently the resuspension was dialysed overnight in 2L Dialysis Buffer (50 mM Tris-HCl, pH 8.0; 0.1 mM EDTA; 5% (v/v) Glycerol; 0.1 mM DTT; 50 mM NaCl). The Dialysis Buffer was exchanged for fresh Dialysis Buffer after at least 2 hours of dialysis. After dialysis the lysate was centrifuged (11,000 g for 20 min at 4 °C). The resulting supernatant contained the required protein which was concentrated to 2 mL using Amicon Ultra-15 centrifugal filters with a 10kDa cut-off (Millipore (UK) Ltd.) and further purified using size exclusion chromatography. The protein was eluted in

Chromatography Buffer and concentrated to 2 mL as previously described. Protein concentration was determined with Quick Start Bradford 1x reagent (Bio-Rad) and adjusted using Chromatography Buffer.

## HSQC NMR

The ES domain, ES$^{388-594}$, protein was analysed by NMR at 5°C under reducing conditions (DTT 10 mM), buffered at pH 8.0 (Tris 20 mM). $^{1}$H-$^{15}$N HSQC was performed at 950 MHz, TCI probe, Bruker following the parameters described in *Figure 1—figure supplement 1*. Concentrations used were 50 μM ES$^{388-594}$ and 500 μM IAA ligand (*i.e.* 1:10).

## Isothermal titration calorimetry (ITC)

ITC was carried out on a MicroCal PEAQ-ITC (Malvern) at 25°C in a Buffer A (sodium phosphate 20 mM, pH 8.0; NaCl 500 mM). Ligand (2 mM IAA) was injected (19 × 4.0 μl) at 150 s intervals into the stirred (500 rpm) calorimeter cell (volume 270 μl) containing 50 μM ES$^{388-594}$ protein. Titration of Buffer A into 50 μM ES$^{388-594}$ protein and IAA (2 mM) into Buffer A served as negative controls. Measurements of the binding affinity of all the titration data were analysed using the MicroCal Software (Malvern).

## Accessions

ETT, AT2G33860; TPL, AT1G15750; TPR1, AT1G80490, TPR2, AT3G16830; TPR3, AT5G27030; TPR4, AT3G15880; HDA6, AT5G63110; HDA19, AT4G38130; HEC1, AT5G67060; PID, AT2G34650; WUS, AT2G17950.

## Acknowledgements

We are grateful to Yuli Ding, Yang Dong, Emilie Knight, Mikhaela Neequaye, Nicola Stacey, Sophia Stavnstrup, Billy Tasker-Brown for critical comments on the manuscript, to Keiko Torii and Shinya Hagihara for the ccvTIR1 line and cvxIAA ligand, to Rebecca Mosher for assistance with the phylogenetic analysis of ETT protein sequences, to Thomas Laux for *TPL::TPL-GFP* and *35S::HDA19-GFP* lines and to Salomé Prat for *35S::TPL-HA* construct. Our gratitude also goes to Felix Ramos-León, Kelley Gallagher and Mark Buttner for assistance with protein production, Clare Stevenson for ITC support, the Norwich Research Park Bioimaging for skillful assistance and the NMR facility in the Astbury Centre, Faculty of Biological Sciences for access to the 950 MHz and 600 MHz spectrometers funded by the University of Leeds. We thank Arnout Kalverda for assistance with analysing the NMR data.

## Additional information

### Funding

| Funder | Grant reference number | Author |
| --- | --- | --- |
| Biotechnology and Biological Sciences Research Council | BB/S002901/1 | Lars Østergaard |
| Biotechnology and Biological Sciences Research Council | BB/L010623/1 | Stefan Kepinski |
| Biotechnology and Biological Sciences Research Council | BB/M011216/1 | André Kuhn |
| Biotechnology and Biological Sciences Research Council | BB/P013511/1 | Lars Østergaard |

The funders had no role in study design, data collection and interpretation, or the decision to submit the work for publication.

### Author contributions

André Kuhn, Data curation, Formal analysis, Validation, Investigation, Visualization, Methodology; Sigurd Ramans Harborough, Heather M McLaughlin, Data curation, Formal analysis, Validation,

Investigation, Methodology; Bhavani Natarajan, Inge Verstraeten, Data curation, Investigation, Writing - review and editing; Jiří Friml, Formal analysis, Writing - review and editing; Stefan Kepinski, Formal analysis, Funding acquisition; Lars Østergaard, Conceptualization, Formal analysis, Supervision, Funding acquisition, Project administration

**Author ORCIDs**
André Kuhn (iD) https://orcid.org/0000-0003-2144-8413
Sigurd Ramans Harborough (iD) https://orcid.org/0000-0002-0361-5647
Heather M McLaughlin (iD) http://orcid.org/0000-0002-3020-7964
Bhavani Natarajan (iD) https://orcid.org/0000-0001-6852-7130
Inge Verstraeten (iD) https://orcid.org/0000-0001-7241-2328
Jiří Friml (iD) http://orcid.org/0000-0002-8302-7596
Stefan Kepinski (iD) https://orcid.org/0000-0001-9819-5034
Lars Østergaard (iD) https://orcid.org/0000-0002-8497-7657

**Decision letter and Author response**
Decision letter https://doi.org/10.7554/eLife.51787.sa1
Author response https://doi.org/10.7554/eLife.51787.sa2

## Additional files

### Supplementary files

• Transparent reporting form

### Data availability

All data generated or analysed during this study are included in the manuscript and supporting files. Source data files have been provided for Figures 1, 2, 3, 4 and 5.

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
