## [Decision Letter]

**Acceptance summary:**

This is an outstanding study that deepens our mechanistic understanding of a non-canonical auxin perception mechanism and we are very pleased that *eLife* can provide a forum for this exciting set of data.

**Decision letter after peer review:**

Thank you for submitting your article "Direct ETTIN-auxin interaction controls chromatin state in gynoecium development" for consideration by *eLife*. Your article has been reviewed by two peer reviewers, and the evaluation has been overseen by a Reviewing Editor and Christian Hardtke as the Senior Editor. The reviewers have opted to remain anonymous.

The reviewers have discussed the reviews with one another and the Reviewing Editor has drafted this decision to help you prepare a revised submission.

While both reviewers highlighted the suitability of your work for *eLife* in principle, they also see certain shortcomings, which prevents publication in its present form.

Essential revisions include:

1) The reviewers see a big gap between the in vitro ETT/TPL interaction and the H3K27ac changes in vivo that needs to be bridged. TPL is not the only co-repressor complex that may recruit *HDA19* to ETT and ETT interacts with other transcription factors that in turn can also recruit co-repressors. It is currently unclear whether auxin-mediated disruption of the ETT/TPL interaction directly induces changes in *HDA19* interaction or chromatin recruitment. This should be tested directly in the presence of auxin in transient interaction assays and by ETT/TPL/HDA19 ChIP +/- IAA at target loci (see further comments to this aspect by reviewer #1).

2) The reviewers also request additional experiments, consolidating the auxin binding to ETT. They encourage the authors to carry out additional MST analyses with ideally using full length ETTIN vs ettin mutant protein (W505). Please quantify binding ETT binding to auxin (see further comments to this aspect by reviewer #2).

Please, see also additional comments by the reviewers below, which may guide you to further strengthen your manuscript.

Reviewer #1:

This study picks up where Simonini et al., 2016 had left off, which had showed that multiple TFs including IND interact with ETT in an auxin -dependent manner. ETT-IND were found to repress *PID* expression based on in situ hybridization in mutant gynoecia. Mutations in IND that disrupted interaction with ETT led to reduced accumulation of *PID*, which was interpreted as a role in transcriptional activation of *PID* by IND and ETT in the absence of auxin.

As I am not an expert on HSQC-NMR or ITC, I will let other reviewers comment on these data. The authors show that nicely TIR1 is neither necessary nor sufficient for *HEC1* and *PID* are induction, suggesting that Aux/IAA degradation is not required for this processes. Critiques that should be addressed are in the numbered/lettered paragraphs below.

1) They next test physical interaction between ETT and TPL/TPRs in Y2H and show it is abolished by addition of IAA. They also test interaction by co-IP in 35S:OE lines after transient transfection into N. benthamiana. In plants, the IAA treatment seems to lead to a both strong reduction in TPL accumulation as well as interaction. This needs to be addressed. It is moreover critical to assess the *HDA19* interaction with ETT in N. benthamiana. Using ETT and *HDA19* co-transfection into plant cells, ETT was recently shown to interact with *HDA19* via endogenous co-repressors (such as TPL or SEU) by co-IP and by BiFC (Chung et al., 2019). In addition, the ETT interaction partner FIL also interacts with *HDA19* (likely via different co-repressors). So it is critical to assess whether auxin disrupts ETT complex formation with *HDA19* to show that the auxin disrupted ETT/TPL interaction is important for *HDA19* association with ETT. As an aside, it should be mentioned in the text that ETT/FIL were shown to recruit *HDA19* to *STM* in the above-mentioned study (Chung et al., 2019).

In ett or tpl /tpr mutant gynoecioa, auxin was not able to induce *HEC1* and *PID*, this was true to a lesser degree for *HEC1* in the *hda19* mutant. In addition, *HEC1* and *PID* expression is elevated in ett, tpl and *hda19* mutants relative to wt. Although not based on inducible mutants, these data suggest that in the wild type ETT and TPL/R repress *HEC1* and IND.

2) Figure 5 attempts to show a possible mechanisms for the observed effects by using chromatin immunoprecipitation of ETT, *HEC1* and *HDA19*. There are several issues with the data provided.

a) First, from a technical perspective, the authors should show the factor binding as% input (to give the reader an idea of the occupancy compared to other TFs or chromatin regulators). More importantly, H3K27ac should be presented as fraction of H3. The latter is critical as a change in histone modification is only informative if we know that the nucleosome occupancy has not changed.

b) Second, there was no test for a possible change in TFL or *HDA19* occupancy upon auxin treatment at *PID* or *HEC1*, although this is a key tenet of their model. ETT occupancy at *PID* and *HEC1* should also be assayed plus minus auxin.

Third, the authors reported lower H3K27ac in the non auxin-treated wild type. This fits with the reduced gene expression, yet cause and consequence are not clear – especially in the absence of test of disruption of ETT- HDA19 interaction by auxin or *HDA19* occupancy at the loci being tested. This is important, because changes in gene expression lead to changes in histone modifications and vice -versa.

3) Finally, the title and main conclusion are misleading and oversell the findings.

a) According to the author's data and with some additional support as specified above, the manuscript could show that ETT serves to recruit TPL and *HDA19* in the absence of IAA. In the presence of IAA this complex may fall apart. This does not show that ETT toggles between activation and repression. It would show that ETT 's ability to recruit co-repressors is abrogated by auxin.

b) I know the authors suggested this activator/repressor model in the Simonini paper based on dominant mutant proteins, which are tricky to interpret. Besides there being no evidence for such a dual role of ETT in the current paper, I think it is important to remember that the ett loss of function mutant has elevated *PID* and *HEC1*. This suggest ETT acts as a repressor. Likewise, for genes bound by MP where loss of MP function results in loss-of-expression, MP is an activator, even though MP does associate with different types of chromatin regulators in the absence or presence of auxin at these loci.

Reviewer #2:

The authors deal with the idea that ETT is a particular ARF protein, that engages directly not only other transcriptional regulators, but the family of TPL co-repressors, and more importantly binds directly auxin. Two previous manuscripts from the same research group (Simonini et al., 2016 and Simonini et al., 2017) had brought up the very intriguing hypothesis that ETT-binds auxin by a non-canonical mechanism, independently of TIR1/AFB-AUX/IAA auxin perception. The hypothesis is fascinating, and the authors previously used genetics, plant physiology and developmental genetics to validate their claim.

In Kuhn et al., the authors seek to: show biochemically direct auxin binding by ETT; connect the ETT "auxin binding role" to specific non-canonical gene expression; bring evidence of ETT directly binding TPL and TPRs for auxin-triggered responses in the gynoecium; show that ETT interactions with TPL and histone deacetylases are sensitive to auxin levels, regulating thereby *HEC1* and *PID* expression. Kuhn et al. work is good and straightforward, in the sense that their hypotheses were addressed by few, but robust experiments. The manuscript would tremendously profit from additional biochemical or biophysical experiments though, using preferably the full length of ETT, to evidence auxin binding. Is that feasible?

Since the authors identified W505 as important putative auxin binding site, why was not an ETT W505 mutant version, and other mutations, tested in iTC or MST, for instance? This reviewer is concerned that since the interaction between ETT and auxin is weak, one could almost account for random peptide interaction, via charges or the similar, for the poor ITC data. Can the authors please quantify binding?

– Along the same lines and about Figure 1 and interactions experiments: Why did the authors not estimate the binding affinity or Kd of ETT-IAA? could the weak (not saturable) ETT-IAA signal be the result of transient peptide interactions?

– When the authors implemented the synthetic auxin-TIR1 tool developed in Keiko Torii's lab, they showed that ETT interaction is highly specific for IAA, but not cvxIAA even though the interaction occurs in an IDR. Can the authors please explain the findings?

– Have the authors evaluated whether there is any sequence conservation for IAA binding around ETT W505?

– Can the authors make the binding comparison of ETT-IAA vs. ETT-BA, or vs. ETT-other auxins, or ETT-tryptophan via iTC in order to make a more robust claim about auxin binding?

– Regarding Model Figure 5D. Do the authors have any hypothesis about what happens with ETT upon auxin binding?

Specific comments:

– The Abstract includes references, is that not a bit unusual?

– Abstract fifth sentence: Sentence too long. Split and/or rephrase

– Abstract final sentence: Giving the outcome of the paper, I recommend to state "ETT might be able to switch chromatin locally…”

– Introduction first paragraph: The word "ubiquitinylation" is incorrect, as it does not refer to ubiquitin conjugation of proteins. Please use "ubiquitination" or "ubiquitylation" instead

– Please rephrase the statement about ES domain being disordered, as it is disconnected, and the reason for bringing it up on that section of the Introduction is not clear.

– The authors used the word "sensitivity", which is a broad claim, given they used a single IAA concentration of 100 µM in their yeast-two hybrid assay.

– HSQC is not the typical experimental approach to proof an X ligand binds to a putative receptor. If this approach is implemented because of the solely disordered nature of ETT, please say it so or rephrase.

– "Helical character" indicated by what? It would be nice to have a visualization help for non-NMR experts.

– “…found that a number of residues shifted…". Please elaborate.

– Why did the authors use benzoic acid (BA) as a control ligand in their experiments? Why not to use tryptophan as the obvious non-active auxin precursor?

– "The HSQC experiment demonstrates that ETT binds IAA directly". I recommend the authors to be cautious here with this statement. While high-field NMR is a powerful tool for investigating transiently forming protein-ligand complexes and for the identification of protein binding partners, there are a number of strategies that the authors must implement first that will specifically enable the physicochemical characterization of ETT-IAA interactions.

– iTC experiments are unfortunately weak e.g. no saturation reached, extremely low signal to noise ratio. Therefore, the authors again should be careful about their statements on direct biochemically relevant ETT-IAA binding.

– Please specify if the ccvTIR1 can accommodate the native IAA.

– Please use here "…independently of the canonical TIR1/AFB1-5 signalling pathway."

– Why IAA = auxin? This is unnecessary and not accurate, stick to "IAA" please.

– Could the authors elaborate on a hypothesis for the role of TPR4? Could its action and interaction with ETT or an ETT-like protein in a different tissue be also auxin sensitive?

– "low levels of auxin" do not maintain per se ETT-TPL/TPR2 associations. Please rephrase.

– The authors have referred twice about the "instantly reversible" event. What do they mean by this? There is no data on timing! "Instantly" as when a [IAA] threshold has been reached?

– The inclusion of an auxin insensitive ett mutant might have shed light on the true nature of the ETT-IAA responsive genes.

Comments on Figures:

– Figure 1. Which IAA concentration was used for these experiments? Please include relevant info in the Figure legend.

– Figure 1. The color of NMR data is hard to distinguish, particularly the blues and the blacks.

Figure 1D-F. Are the control panels E and F necessary as extra figure? Cannot they be included in the D?

– Figure 2. Colors in panel d are difficult to be distinguished. Please recolor.

– Figure 3A. Please improve, resize and label residues. Mark and depict W505.

– Figure 3B. What is WTL? Give info in Legend, please.

[Editors' note: further revisions were suggested prior to acceptance, as described below.]

Thank you for submitting your article "Direct ETTIN-auxin interaction controls chromatin states in gynoecium development" for consideration by *eLife*. Your article has been reviewed by one peer reviewers, and the evaluation has been overseen by Reviewing Editor Jürgen Kleine-Vehn and Christian Hardtke as the Senior Editor The reviewers have opted to remain anonymous.

The reviewers have discussed the reviews with one another and the Reviewing Editor has drafted this decision to help you prepare a revised submission.

We would like to draw your attention to changes in our policy on revisions we have made in response to COVID-19 (https://elifesciences.org/articles/57162). Specifically, we are asking editors to accept without delay manuscripts, like yours, that they judge can stand as *eLife* papers without additional data, even if they feel that they would make the manuscript stronger. Thus the revisions requested below only address clarity and presentation.

Both reviewers agreed that you convincingly revised your manuscript and that your manuscript is in principle ready for publication. Reviewer #1 asks you to modify some of your statements and your overall model depiction (see point 2 below).

Reviewer #1:

The authors have addressed the majority of my concerns. I do not recommend additional experiments, but provide a clarification based on the author response. In addition, I ask for changes in the text and model to address my comments in point 2 below.

Previous review

1) It is moreover critical to assess the *HDA19* interaction with ETT in N. benthamiana. Using ETT and *HDA19* co-transfection into plant cells, ETT was recently shown to interact with *HDA19* via endogenous co-repressors (such as TPL or SEU) by co-IP and by BiFC (Chung et al., 2019). In addition, the ETT interaction partner FIL also interacts with *HDA19* (likely via different co-repressors). So it is critical to assess whether auxin disrupts ETT complex formation with *HDA19* to show that the auxin disrupted ETT/TPL interaction is important for *HDA19* association with ETT.

Authors: We are confused about the first sentence of this section that *HDA19* interaction with ETT in *N. benthamiana* must be assessed. As mentioned, this experiment was recently published in Chung et al., 2019, and showed that *HDA19* and ETT do not interact directly, but via other proteins. Unless we have misunderstood the reviewer's comment, we do not see the need to carry this out again.

Reviewer: *N. benthamiana* assays also probe indirect interactions between proteins via host proteins. Perusal of the pulldowns in Chung et al. does indeed reveal a weak *HDA19* pulldown by ETT (and a very strong ARF4 pulldown by *HDA19*). BiFC using Arabidopsis protoplasts showed strong (indirect) interactions between ETT and *HDA19*. The current study suggests that these indirect interactions should be impaired in the presence of high levels of auxin.

Current review

2) The authors should reconcile their data with the previously published data that implicates ETT/ARF4/FIL in recruitment of *HDA19* containing co-repressor complexes to repress KNOX genes in regions of local auxin maxima at the inflorescence shoot apex. In looking at the literature and the current manuscript, a difference in effective auxin levels became apparent. The authors used a rather high auxin concentration to disrupt the ETT – *HDA19* co-repressor complex in planta: 100 µM IAA for Figure 4 and Figure 5. By contrast 10 µM IAA is commonly used in inflorescences. Gynoecia may indeed have higher auxin levels.

This suggests a different model and a different conclusion from that currently proposed by the authors are needed. At intermediate auxin levels, for example those of the inflorescence shoot apex, ETT (together with other TFs) is able to recruit *HDA19*-containing co-repressors, while at higher auxin levels, those present in the gynoecium, this interaction is blocked. The authors should revise the Discussion and their model accordingly.

---

## [Author Response]

Essential revisions include:

*1) The reviewers see a big gap between the* in vitro *ETT/TPL interaction and the H3K27ac changes* in vivo *that needs to be bridged. TPL is not the only co-repressor complex that may recruit HDA19 to ETT and ETT interacts with other transcription factors that in turn can also recruit co-repressors. It is currently unclear whether auxin-mediated disruption of the ETT/TPL interaction directly induces changes in HDA19 interaction or chromatin recruitment. This should be tested directly in the presence of auxin in transient interaction assays and by ETT/TPL/HDA19 ChIP +/- IAA at target loci (see further comments to this aspect by reviewer #1).*

We agree that this direct connection between ETT/TPL and *HDA19*/H3K27ac was not established in the original submission. We performed the ChIP experiments suggested showing that the interaction of both *HDA19* and TPL with the promoters of *HEC1* and *PID* is disrupted in the presence of IAA, while ETT binding remains (as previously shown – Simonini et al., 2016). These data support that *HDA19* is recruited by ETT via TPL and agree with the disruption of protein interactions between ETT and TPL by IAA. Whilst *HDA19* does not directly interact with ETT (Chung et al., 2019), the data support that *HDA19* forms part of the repressive ETT/TPL complex that is disrupted by IAA. The data are presented in the revised manuscript.

We did not carry out the ETT-*HDA19* interaction assay in *N. benthamiana* but refer to Chung et al., 2019, where our experiments on this are published. Since no interaction was observed, we suggest it may not be that informative to carry this out +/- IAA.

2) The reviewers also request additional experiments, consolidating the auxin binding to ETT. They encourage the authors to carry out additional MST analyses with ideally using full length ETTIN vs ettin mutant protein (W505). Please quantify binding ETT binding to auxin (see further comments to this aspect by reviewer #2).

Over several years, we tried a range of expression systems and hosts to produce full-length ETT for biochemical studies with no success. Additional attempts could take a long time with no guarantee for success. However, we have previously shown that the ES domain is sufficient for the IAA-sensitivity (Simonini et al., 2016; Simonini et al., 2018) in agreement with numerous other studies reporting a biophysical interaction using an individual protein domain in isolation (e.g. Martin-Arevalillo et al. 2017 PNAS 114, 8107-12; Wild et al. (2016) Science 352, 986-90).

We have not been able to improve on the ITC data presented in the first version. This is due to different factors. Firstly, even with a further optimisation of ES-domain production, we are unable to obtain large enough amounts of fully soluble and stable protein. Secondly, the protein we use seems to aggregate during the ITC measurements making it difficult to obtain a K_d_ value. This is now discussed in the revised manuscript.

In the revised manuscript, we instead include an additional independent experiment using the DARTS assays providing additional evidence for the direct interaction between ETT and IAA (presented in Figure 1A).

Finally, we also show that the tryptophan in position 505 (W505) is required for the IAAsensitivity of ETT in both Y2H and transient assays in N. benthamiana.

Reviewer #1:1) They next test physical interaction between ETT and TPL/TPRs in Y2H and show it is abolished by addition of IAA. They also test interaction by co-IP in 35S:OE lines after transient transfection into N. benthamiana. In plants, the IAA treatment seems to lead to a both strong reduction in TPL accumulation as well as interaction. This needs to be addressed.

We agree with the reviewer that the previous figure had lower levels of TPL at higher concentrations of IAA. We have carried this out several times without an effect on TPL levels, but showing the reduction in ETT-TPL interaction. We have now included the results from a more representative experiments (Figure 3C,D, Figure 3—figure supplement 3).

It is moreover critical to assess the HDA19 interaction with ETT in N. benthamiana. Using ETT and HDA19 co-transfection into plant cells, ETT was recently shown to interact with HDA19 via endogenous co-repressors (such as TPL or SEU) by co-IP and by BiFC (Chung et al., 2019). In addition, the ETT interaction partner FIL also interacts with HDA19 (likely via different co-repressors). So it is critical to assess whether auxin disrupts ETT complex formation with HDA19 to show that the auxin disrupted ETT/TPL interaction is important for HDA19 association with ETT. As an aside, it should be mentioned in the text that ETT/FIL were shown to recruit HDA19 to STM in the above-mentioned study (Chung et al., 2019).

As mentioned above, to assess whether auxin disrupts ETT complex formation with *HDA19*, we have performed ChIP with *HDA19*/TPL/ETT +/-IAA and found that the interaction of *HDA19* and TPL with the *HEC1* and *PID* promoters is sensitive to IAA, while ETT remains bound independently of IAA levels. This supports the model that ETT recruits *HDA19* via interactions with TPL and that the ETT/HDA19/TPL complex is disrupted by IAA. These data are included in the manuscript (Figure 5A-C).

We are confused about the first sentence of this section that *HDA19* interaction with ETT in *N. benthamiana* must be assessed. As mentioned, this experiment was recently published in Chung et al., 2019, and showed that *HDA19* and ETT do not interact directly, but via other proteins. Unless we have misunderstood the reviewer’s comment, we do not see the need to carry this out again.

We agree with the reviewer about specifically mentioning the previous finding that ETT is required to recruit *HDA19* to the *STM* promoter as reported in Chung et al., 2019. This is now included in the revised version of the manuscript.

In ett or tpl /tpr mutant gynoecioa, auxin was not able to induce HEC1 and PID, this was true to a lesser degree for HEC1 in the hda19 mutant. In addition, HEC1 and PID expression is elevated in ett, tpl and hda19 mutants relative to wt. Although not based on inducible mutants, these data suggest that in the wild type ETT and TPL/R repress HEC1 and IND.2) Figure 5 attempts to show a possible mechanisms for the observed effects by using chromatin immunoprecipitation of ETT, HEC1 and HDA19. There are several issues with the data provided.a) First, from a technical perspective, the authors should show the factor binding as% input (to give the reader an idea of the occupancy compared to other TFs or chromatin regulators). More importantly, H3K27ac should be presented as fraction of H3. The latter is critical as a change in histone modification is only informative if we know that the nucleosome occupancy has not changed.

We agree with the technical concerns raised by the reviewer and have revised the figure as suggested (Figure 5D, E).

b) Second, there was no test for a possible change in TFL or HDA19 occupancy upon auxin treatment at PID or HEC1, although this is a key tenet of their model. ETT occupancy at PID and HEC1 should also be assayed plus minus auxin.

We agree with this point and have performed additional ChIP experiments plus/minus auxin to test ETT, TPL and *HDA19* occupancy at *PID* and *HEC1*. The new data have been included in the revised figure (Figure 5A-C).

Third, the authors reported lower H3K27ac in the non auxin-treated wild type. This fits with the reduced gene expression, yet cause and consequence are not clear – especially in the absence of test of disruption of ETT- HDA19 interaction by auxin or HDA19 occupancy at the loci being tested. This is important, because changes in gene expression lead to changes in histone modifications and vice -versa.

We agree that this is an important point and have addressed experimentally the *HDA19* occupancy at *PID* and *HEC1* by ChIP as requested by the reviewer above. Moreover, we have inserted a sentence to highlight that no direct interaction between ETT and *HDA19* was detected in both Y2H and Co-IP (Chung et al., 2019); however, an indirect interaction via a co-repressor was suggested.

3) Finally, the title and main conclusion are misleading and oversell the findings.a) According to the author's data and with some additional support as specified above, the manuscript could show that ETT serves to recruit TPL and HDA19 in the absence of IAA. In the presence of IAA this complex may fall apart. This does not show that ETT toggles between activation and repression. It would show that ETT 's ability to recruit co-repressors is abrogated by auxin.

We do not agree that the title is misleading and would prefer to keep as is. However, we have modified the concluding remarks to take the reviewer’s comments on repressive/de-repressive ability of ETT on board. This has led to a section in which we suggest the possibility that auxin either renders ETT a passive DNA-binding protein or has a role in recruiting activating components.

b) I know the authors suggested this activator/repressor model in the Simonini paper based on dominant mutant proteins, which are tricky to interpret. Besides there being no evidence for such a dual role of ETT in the current paper, I think it is important to remember that the ett loss of function mutant has elevated PID and HEC1. This suggest ETT acts as a repressor. Likewise, for genes bound by MP where loss of MP function results in loss-of-expression, MP is an activator, even though MP does associate with different types of chromatin regulators in the absence or presence of auxin at these loci.

As above to 3a.

Reviewer #2:[…] In Kuhn et al., the authors seek to: show biochemically direct auxin binding by ETT; connect the ETT "auxin binding role" to specific non-canonical gene expression; bring evidence of ETT directly binding TPL and TPRs for auxin-triggered responses in the gynoecium; show that ETT interactions with TPL and histone deacetylases are sensitive to auxin levels, regulating thereby HEC1 and PID expression. Kuhn et al. work is good and straightforward, in the sense that their hypotheses were addressed by few, but robust experiments. The manuscript would tremendously profit from additional biochemical or biophysical experiments though, using preferably the full length of ETT, to evidence auxin binding. Is that feasible?

Despite using a large range of different expression systems over several years, we are unable to produce full length ETT in any recombinant system. As described in Simonini et al., 2018, we can make an almost full version of the Cterminal ES domain (only lacking 14 amino acids at the terminus), which is sufficient for the ETT-sensitivity.

Since the authors identified W505 as important putative auxin binding site, why was not an ETT W505 mutant version, and other mutations, tested in iTC or MST, for instance? This reviewer is concerned that since the interaction between ETT and auxin is weak, one could almost account for random peptide interaction, via charges or the similar, for the poor ITC data. Can the authors please quantify binding?

As explained above, we have not been able to improve on our ITC data despite several attempts (including developing a fusion protein that stayed in solution, but had lost ability to bind IAA in this assay). However, we agree with the reviewer on exploiting the potential importance of the tryptophan in position 505 (W505) in the ETT-auxin interaction. Therefore, we developed mutant versions substituting tryptophan for alanine (W505A) and tested them in interaction assays ± IAA (Y2H and Co-IP with TPL). In agreement with a central role of W505, the ETT(W505A)-TPL interaction was not disrupted by IAA (Figure 3B, D).

– Along the same lines and about Figure 1 and interactions experiments: Why did the authors not estimate the binding affinity or Kd of ETT-IAA? could the weak (not saturable) ETT-IAA signal be the result of transient peptide interactions?

Possibly due to insufficiently high protein concentration and/or aggregation, we were unable to obtain ITC data allowing quantification of the binding affinity.

– When the authors implemented the synthetic auxin-TIR1 tool developed in Keiko Torii's lab, they showed that ETT interaction is highly specific for IAA, but not cvxIAA even though the interaction occurs in an IDR. Can the authors please explain the findings?

Without a protein structure it is hard to explain the specific requirement for IAA over other auxinic compounds in this pathway. However, this is not just IAA vs cvxIAA. We have previously reported that neither NAA nor 2,4-D are able to mediate the effect on ETTprotein interactions (this paper and Simonini et al., 2016).

– Have the authors evaluated whether there is any sequence conservation for IAA binding around ETT W505?

This is an excellent point. We have found that the tryptophan in position 505 is highly conserved among the 25 ETT proteins that we have identified. In contrast, very little sequence conservation is observed among other residues in that region. We have included a sequence logo of the W505 region in the alignment in Figure 3—figure supplement 1A to point this out.

– Can the authors make the binding comparison of ETT-IAA vs. ETT-BA, or vs. ETT-other auxins, or ETT-tryptophan via iTC in order to make a more robust claim about auxin binding?

In the revised manuscript, we use IAA vs BA in the DARTS and NMR experiments and IAA vs. NAA in the Y2H and Co-IP in N. benthamiana. We did not test all variations in ITC due to difficulty in obtaining sufficient protein as described above.

– Regarding Model Figure 5D. Do the authors have any hypothesis about what happens with ETT upon auxin binding?

Structurally, we do not yet have any idea what changes are occurring in the ETT protein upon IAA binding, although, our recent data included in this revised manuscript suggest that the tryptophan in position 505 may play an important role. Moreover, in response to the comments from reviewer 1, we have modified the speculations to include the possibility that ETT functions primarily as a repressor and that IAA binding induces a conformational change in ETT to make it unable to bind the co-repressive complex. An alternative hypothesis is that IAA-bound ETT is required for de-repression by recruiting activators of gene expression. Both possibilities are presented in the revised version.

Specific comments:– The Abstract includes references, is that not a bit unusual?

We are happy to exclude if the journal requests this. However, it seems some *eLife* papers have references in the Abstract whereas others do not.

– Abstract fifth sentence: Sentence too long. Split and/or rephrase

We agree and have modified the sentence.

– Abstract final sentence: Giving the outcome of the paper, I recommend to state "ETT might be able to switch chromatin locally…”

We have modified this sentence.

– Introduction first paragraph: The word "ubiquitinylation" is incorrect, as it does not refer to ubiquitin conjugation of proteins. Please use "ubiquitination" or "ubiquitylation" instead

This has been corrected.

– Please rephrase the statement about ES domain being disordered, as it is disconnected, and the reason for bringing it up on that section of the Introduction is not clear.

We agree and have modified the sentence.

– The authors used the word "sensitivity", which is a broad claim, given they used a single IAA concentration of 100 µM in their yeast-two hybrid assay.

We disagree and prefer to use the term ‘sensitivity’. We have previously shown that the interactions of ETT with the IND TF occurs with several IAA concentrations (Simonini et al., 2016).

– HSQC is not the typical experimental approach to proof an X ligand binds to a putative receptor. If this approach is implemented because of the solely disordered nature of ETT, please say it so or rephrase.

NMR was used to investigate the binding of auxin to ETT principally because it is a powerful technique for analysing interactions that are relatively low affinity, as was suspected in this case. Such HSQC experiments examine the interaction from the perspective of the receptor rather than the ligand (as is the case in WaterLOGSY NMR experiments) and provide information on the number of residues of the receptor that are affected by the binding of the ligand, indicated by changes in the position of HN cross peaks within the spectra.

– "Helical character" indicated by what? It would be nice to have a visualization help for non-NMR experts.

The helical character is typically indicated by the dispersed HN cross peaks between 7.8 to 7.0 ppm on the 1H chemical shift axis of the HSQC spectrum. This has now been indicated in Figure 1B, C.

– “…found that a number of residues shifted…". Please elaborate.

Perturbations in the position of the HN cross peaks in the HSQC is due to changes in the local chemical environment of the amide bonds within the protein. This is caused either by being in close proximity to bound ligand and/or due to conformational change to the protein induced by the binding event. These IAA specific induced changes are shown in Figure 1D.

– Why did the authors use benzoic acid (BA) as a control ligand in their experiments? Why not to use tryptophan as the obvious non-active auxin precursor?

Benzoic acid was selected as the control because it has no auxinic activity but like IAA, is a weak acid.

– "The HSQC experiment demonstrates that ETT binds IAA directly". I recommend the authors to be cautious here with this statement. While high-field NMR is a powerful tool for investigating transiently forming protein-ligand complexes and for the identification of protein binding partners, there are a number of strategies that the authors must implement first that will specifically enable the physicochemical characterization of ETT-IAA interactions.

The IAA specific induced changes in the HSQC spectrum is the effect of IAA binding directly to ETT, either by direct interaction with the binding interface or due to more distal conformational changes as a result of binding. This protein-ligand complex is further supported by ITC and DARTS assays. These results do not rule out possibility that the bioactive complex involves further binding partners.

– iTC experiments are unfortunately weak e.g. no saturation reached, extremely low signal to noise ratio. Therefore, the authors again should be careful about their statements on direct biochemically relevant ETT-IAA binding.

We agree with the reviewer and have included text to explain the problems we have experienced with the ITC. In addition to the HSQC-NMR data we now also show data from a DARTS experiment that provide additional evidence of direct interaction (Figure 1A).

– Please specify if the ccvTIR1 can accommodate the native IAA.

This is described in Uchida et al., 2018 and their data show that ccvTIR1 does not respond to IAA. This is now specifically mentioned in the manuscript.

– Please use here "…independently of the canonical TIR1/AFB1-5 signalling pathway."

This has been modified.

– Why IAA = auxin? This is unnecessary and not accurate, stick to "IAA" please.We have changed to (primarily) use IAA and have specified this earlier in manuscript.– Could the authors elaborate on a hypothesis for the role of TPR4? Could its action and interaction with ETT or an ETT-like protein in a different tissue be also auxin sensitive?

This is an entirely possible scenario and we have included a sentence in the revised manuscript to that extent.

– "low levels of auxin" do not maintain per se ETT-TPL/TPR2 associations. Please rephrase.

We agree and have modified this sentence.

– The authors have referred twice about the "instantly reversible" event. What do they mean by this? There is no data on timing! "Instantly" as when a [IAA] threshold has been reached?

We have modified the text at both places in response to the reviewer’s comment. In fact, we like the ‘threshold’ formulation so much that we have included this in the revised manuscript.

– The inclusion of an auxin insensitive ett mutant might have shed light on the true nature of the ETT-IAA responsive genes.

We are not entirely sure what the reviewer means by “the true nature of the

ETT-IAA responsive genes”; however, we refer to the paper by Simonini et al., 2016, where expressing a version of the TF IND (IND^D30G^) that makes the ETT-IND interaction insensitive leads to constitutive repression of *PID* (shown by in situ hybridisation in the gynoecium).

Comments on Figures:– Figure 1. Which IAA concentration was used for these experiments? Please include relevant info in the Figure legend.

ES^388-594^:IAA / 1:10 / 50 µM:500 µM. This has now been indicated in the Figure legend.

– Figure 1. The color of NMR data is hard to distinguish, particularly the blues and the blacks.

Colour of the NMR data has been changed to emphasise the contrast with black ES^388-594^ only control sample.

Figure 1D-F. Are the control panels E and F necessary as extra figure? Cannot they be included in the D?

We prefer to include the ITC controls as they are to clearly show the difference between them and the result with both ES and IAA.

– Figure 2. Colors in panel d are difficult to be distinguished. Please recolor.

This has been modified.

– Figure 3A. Please improve, resize and label residues. Mark and depict W505.

This has been modified.

– Figure 3B. What is WTL? Give info in Legend, please.

We believe the reviewer is referring to the -W-L-A-H, which indicates Yeast Selection medium lacking Tryptophan (W), Leucin (L), Adenine (A) and Histidine (H) to assess interaction. This is now specified in the legend.

[Editors' note: further revisions were suggested prior to acceptance, as described below.]

Reviewer #1:The authors have addressed the majority of my concerns. I do not recommend additional experiments, but provide a clarification based on the author response. […]Reviewer: N. benthamiana assays also probe indirect interactions between proteins via host proteins. Perusal of the pulldowns in Chung et al. does indeed reveal a weak HDA19 pulldown by ETT (and a very strong ARF4 pulldown by HDA19). BiFC using Arabidopsis protoplasts showed strong (indirect) interactions between ETT and HDA19. The current study suggests that these indirect interactions should be impaired in the presence of high levels of auxin.

We thank the reviewer for clarifying this comment. We believe the new ChIP data provided using a *p35S:HDA19:GFP* line in presence of auxin also addresses this point showing that auxin leads to dissociation of *HDA19* from ETT-target loci, while ETT remains bound.

Current review2) The authors should reconcile their data with the previously published data that implicates ETT/ARF4/FIL in recruitment of HDA19 containing co-repressor complexes to repress KNOX genes in regions of local auxin maxima at the inflorescence shoot apex. In looking at the literature and the current manuscript, a difference in effective auxin levels became apparent. The authors used a rather high auxin concentration to disrupt the ETT – HDA19 co-repressor complex in planta: 100 µM IAA for Figure 4 and Figure 5. By contrast 10 µM IAA is commonly used in inflorescences. Gynoecia may indeed have higher auxin levels.This suggests a different model and a different conclusion from that currently proposed by the authors are needed. At intermediate auxin levels, for example those of the inflorescence shoot apex, ETT (together with other TFs) is able to recruit HDA19-containing co-repressors, while at higher auxin levels, those present in the gynoecium, this interaction is blocked. The authors should revise the Discussion and their model accordingly.

This is an excellent point and we agree with the reviewer that different auxin concentrations may have different effects in diverse developmental contexts. It is, moreover, possible that different ETT-containing complexes require different auxin concentration thresholds to generate auxin responses. We have modified the Discussion accordingly (first paragraph); however, we think that Figure 5F depicts our model appropriately.